# Ozone and water vapor variability in the polar middle atmosphere observed with ground-based microwave radiometers

Guochun Shi[1], Witali Krochin[1], Eric Sauvageat[1], and Gunter Stober[1]

[1]Institute of Applied Physics & Oeschger Center for Climate Change Research, University of Bern, Bern, Switzerland

**Correspondence:** Guochun Shi (guochun.shi@unibe.ch)

**Abstract.** Leveraging continuous ozone and water vapor measurements with the two ground-based radiometers GROMOS-C and MIAWARA-C at Ny-Ålesund, Svalbard (79°N, 12°E) that started in September 2015 and combining MERRA-2, and Aura-MLS datasets, we analyze the interannual behavior and differences of ozone and water vapor and compile climatologies of both trace gases describing the annual variation of ozone and water vapor at polar latitudes. A climatological comparison of the measurements from our ground-based radiometers with reanalysis and satellite data was performed. Overall differences between GROMOS-C and Aura-MLS ozone volume mixing ratio (VMR) climatology are mainly within $\pm7\%$ throughout the middle and upper stratosphere and exceed 10% in the lower mesosphere (1-0.1 hPa) in March and October. For the water vapor climatology, the average 5% agreement is between MIAWARA-C and Aura-MLS water vapor VMR values throughout the stratosphere and mesosphere (100–0.01 hPa). The comparison to MERRA-2 yields an agreement that reveals discrepancies larger than $50\,\%$ above 0.2 hPa depending on the implemented radiative transfer schemes and other model physics. Furthermore, we perform a conjugate latitude comparison by defining a virtual station in the southern hemisphere at the geographic coordinate (79°S, 12°E) to investigate interhemispheric differences in the atmospheric compositions. Both trace gases show much more pronounced interannual and seasonal variability in the northern hemisphere than in the southern hemisphere. We estimate the effective water vapor transport vertical velocities corresponding to upwelling and downwelling periods driven by the residual circulation. In the northern hemisphere, the water vapor ascent rate (05 May to 20 Jun in 2015, 2016, 2017, 2018 and 2021, and 15 Apr to 31 May in 2019 and 2020) is $3.4\pm1.9$ mm s$^{-1}$ from MIAWARA-C and $4.6\pm1.8$ mm s$^{-1}$ from Aura-MLS, and descent rate (15 Sep to 31 Oct in 2015-2021) is $5.0\pm1.1$ mm s$^{-1}$ from MIAWARA-C and $5.4\pm1.5$ mm s$^{-1}$ from Aura-MLS at the altitude range of about 50-70 km. The water vapor ascent (15 Oct to 30 Nov in 2015-2021) and descent rates (15 Mar to 30 Apr in 2015-2021) in the southern hemisphere are $5.2\pm0.8$ mm s$^{-1}$ and $2.6\pm1.4$ mm s$^{-1}$ from Aura-MLS, respectively. The water vapor transport vertical velocities analysis further reveals a higher variability in the northern hemisphere and is suitable to monitor and characterize the evolution of the northern and southern polar dynamics linked to the polar vortex as a function of time and altitude.

## 1 Introduction

Ozone and water vapor are essential climate variables that play a key role in the radiative balance in the middle atmosphere. Their seasonal and interannual variability is closely coupled to dynamical and chemical processes, which are driven and modu-

lated by atmospheric waves including planetary waves, gravity waves (GWs), and atmospheric tides. These atmospheric waves transport energy and momentum from their source region to the altitudes of their dissipation and, thus, contribute to the energy balance between different atmospheric layers.

Model results suggest that GWs drive the summer mesopause temperature up to 100 K below the radiative equilibrium (Lindzen, 1981; Smith, 2012; Becker, 2012). The extreme cold temperatures at the summer mesopause are the result of an upwelling, and a corresponding adiabatic cooling of the uplifted air masses, in the summer hemisphere and are accompanied by a downwelling in the winter hemisphere. The pole-to-pole circulation is often referred to as residual circulation or transformed Eulerian mean circulation (Andrews and Mcintyre, 1976). Another important circulation branch at altitudes lower than the residual circulation is the Brewer-Dobson Circulation (BDC) (Brewer, 1949; Dobson, 1956). BDC is a factor in stratospheric ozone and water vapor variability but, as mentioned later in the manuscript, polar stratospheric clouds can have significant seasonal impacts on ozone and water vapor abundances in the lower stratosphere. The circulation is fundamentally driven by dissipating waves of tropospheric origin and broadly consists of large-scale tropical ascent and winter pole descent. BDC is much weaker during boreal summer due to the different distribution of land masses and the associated differences in the generation of planetary and GWs between both hemispheres. BDC can govern the entry and distribution of air masses and constituents from the troposphere into and within the stratosphere. The meridional transport of trace gases into the polar cap is controlled by the strength of the polar vortex during polar winter, which is driven by the temperature gradient between the polar cap and the mid-latitudes through the thermal wind balance in the hemispheric winter stratosphere, and forms an essential barrier separating ozone rich air at the mid-latitudes from ozone depleted air within the polar cap. However, planetary waves can disturb the polar vortex and even lead to its breakdown during sudden stratospheric warming events (SSWs) (Matsuno, 1971; Baldwin et al., 2021), which is accompanied by a large-scale intrusion and mixing of air masses from the mid-latitudes towards the high-latitudes helping to recover the ozone VMR (Schranz et al., 2020). Furthermore, the transition from the winter to the summer circulation is decisively controlled by the presence of the planetary wave activities (Matthias et al., 2021). Previous studies even concluded that dynamical forced transitions have a persistent impact on the circulation lasting several weeks (Baldwin and Dunkerton, 2001). The stratospheric quasi-biennial oscillation (QBO) modulates the Northern Hemisphere wintertime stratospheric polar vortex, resulting in its weakening and shifting (Garfinkel et al., 2012; Zhang et al., 2019). Wang et al. (2022) uses reanalysis data and model simulations to demonstrate that the total column ozone and stratospheric ozone (50–10 hPa) anomalies are seasonally dependent and zonally asymmetric in the polar region, which the QBO affects the polar vortex and stratospheric ozone mainly by modifying the wave number 1 activities. Tao et al. (2019) investigate an intercomparison of simulated stratospheric water vapor variations, focusing on the QBO and long-term variability and trends.

Stratospheric ozone observation results largely reflect the distribution that presents significant asymmetry in both hemispheres, with the differences maximizing in the winter and spring seasons (Shepherd, 2008). Because most ozone is found in the lower stratosphere, the differences in the column ozone distribution explain the asymmetry because of dynamic transport, as well as the interannual variability of ozone in both hemispheres (McConnell and Jin, 2008; Langematz, 2019). Long-term polar ozone observations offer a better recognition and predictability of stratospheric ozone trends and an understanding of the attribution of changes. Water vapor has a chemical lifetime of the order of months in the upper stratosphere and lower mesosphere (Brasseur

and Solomon, 2005), therefore, it can be used as a tracer to study a large-scale upwelling and downwelling of the air masses in the polar mesosphere. The mesosphere at the high latitudes is characterized by an annual variation with higher water vapor during local summer and lower water vapor during local winter mainly determined by the mean vertical transport (Forkman et al., 2005; Lee et al., 2011). Straub et al. (2010) and Schranz et al. (2019) estimate the vertical gradient of water vapor inside of the polar vortex in autumn based on microwave radiometry measurements at polar latitudes. The distribution and variability of ozone and water vapor exhibit a wealth of information on atmospheric circulation.

There are several techniques to obtain ozone and water vapor measurements in the middle atmosphere. The Aura satellite with the Microwave Limb Sounder (MLS) collects global water vapor and ozone profiles among other chemical species with coverage at a fixed local time due to its sun-synchronous orbit (Livesey et al., 2006). Ground-based observations are often performed using Brewer and Dobson instruments (Zuber et al., 2021), which provide very high quality and precision ozone column densities, but lack the vertically-resolved information. Lidars are providing good vertical resolution to measure ozone (Brinksma et al., 1997; Bernet et al., 2021). The instruments carried with aircraft and balloon-borne instruments including ozonesondes and frost-point hygrometers perform highly vertically resolved measurements of ozone and water vapor in the upper troposphere and lower stratosphere (Zahn et al., 2014; Eckstein et al., 2017). However, there are only a few systems available and the observation time depends on tropospheric weather conditions. At tropospheric altitudes, water vapor can also be retrieved leveraging Raman-lidars (Sica and Haefele, 2015, 2016). Precise water vapor measurements above the troposphere can also be collected by in-situ balloon-borne sensors such as laser absorption spectrometers (Graf et al., 2021). Ground-based microwave radiometers (MWRs) allow continuous observations under all weather conditions with a time resolution of the order of hours except during rain. MWRs measuring ozone and water vapor are valuable as they complement satellite measurements, are relatively easy to maintain, and have long lifetimes which ensure long and continous time series covering several decades, and can be operated from different locations with measurements performed autonomously on a campaign basis (Scheiben et al., 2013, 2014). Ground-based microwave radiometry is a reliable technique that performs continuous measurements to monitor the vertical profiles of ozone and water vapor VMR changes to investigate Arctic/Antarctic dynamics from diurnal to interannual timescales.

Here, we present a detailed comparison of ozone and water vapor observed by Aura-MLS at conjugate latitude station leveraging multiyear ground-based observations from GROMOS-C and MIAWARA-C performed at Ny-Ålesund and Aura-MLS data as well as reanalysis data. We produce and compare the polar regions' multiyear-mean ozone and water vapor climatologies at conjugate latitude stations (Fig. 1). On one hand, it is intended to provide a well-characterized representation of ozone and water vapor measured by the two instruments and the chemistry differences between both hemispheres concerning climatological behaviors. On the other hand, it provides a source of data for future work including intercomparison studies and evaluation. Furthermore, we use the water vapor mixing ratios measurements from MIAWARA-C and Aura-MLS observation data to derive the ascent and descent rates. We estimate the strength of upwelling and downwelling in both hemispheres over the polar stations and discuss their interannual variability as well as the hemispheric differences.

We provide an overview of the datasets in section 2. The time series of ozone and water vapor at conjugate latitude stations in the Northern Hemisphere (NH) and Southern Hemisphere (SH) are presented in section 3. The climatologies of ozone and

water vapor are discussed in section 4. The transport of water vapor is discussed in section 5. Sections 6 and 7 present the discussion and conclusions of this study.

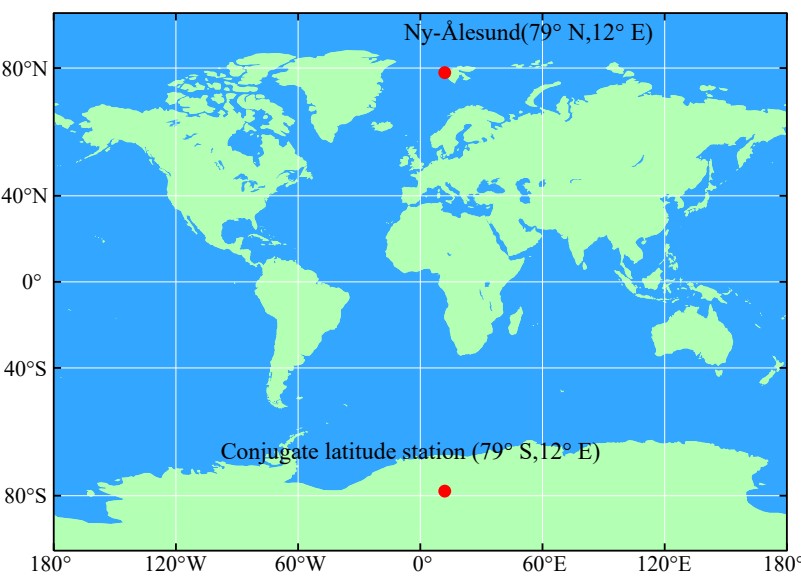

**Figure 1.** The geographical map indicates the two stations in the northern and southern polar regions. The conjugate latitude station in SH is a virtual station for this study. The locations of the stations are indicated with a solid red circle.

## 2 Instruments and models

In this study, we use ozone and water vapor measurements from our two ground-based MWRs GROMOS-C and MIAWARA-C which are only available to measure at single locations and, thus, are representative of a specific geographic location. Both instruments are located at Ny-Ålesund, Svalbard (79° N, 12° E) and collected continuous data since September 2015. We extract the interannual ozone and water vapor variability between Jan 2015 and July 2022 from MERRA-2 and Aura-MLS over northern and southern polar stations. The corresponding virtual conjugate latitude station (79° S, 12° E) is shown in Fig. 1. Additionally, we use temperature observations from Aura-MLS.

### 2.1 GROMOS-C

GROMOS-C (GRound-based Ozone MOnitoring System for Campaigns) is an ozone MWR measuring the ozone emission line at 110.836 GHz at Ny-Ålesund, Svalbard (79° N, 12° E) which is described in detail in Fernández et al. (2015). It was built by the Institute of Applied Physics (IAP) at the University of Bern. GROMOS-C is very compact so it can be transported and operated at remote field sites under extreme climate conditions. It further can switch the frequency of the local oscillator and measure the 115 GHz carbon monoxide (CO) emission line. The system noise temperature of the instrument is about 1080 K. The optics of GROMOS-C has two rotating mirrors such that observations in all four cardinal directions are possible. Therefore,

GROMOS-C observes subsequently on the four cardinal directions (north, east, south, and west) under an elevation angle of 22° with a sampling time of 4 s. Ozone VMR profiles are retrieved from the ozone spectra with a temporal averaging of 2 hours leveraging Atmospheric Radiative Transfer Simulator version-2 (ARTS2; Eriksson et al., 2011) and Qpack2 software (Eriksson et al., 2005) according to the optimal estimation algorithm (Rodgers, 2000). An a priori ozone profile is required for optimal estimation and is taken from an MLS climatology of the years 2004-2013. The sensitive altitude range for this instrument extends from 23 km to 70 km. The vertical resolution of ozone profiles is 10-12 km in the stratosphere and increases up to 20 km in the mesosphere as estimated from the width of the averaging kernels. The averaging kernels (AVKs) of GROMOS-C together with its measurement response, errors, and ozone profiles are shown in Appendix A (Fig. A1). In the lower stratosphere, the errors are below 0.3 ppmv and reach above the stratopause values up to 0.4 ppmv. More details about the uncertainty and the AVKs can be found in Fernández et al. (2015).

## 2.2 MIAWARA-C

MIAWARA-C (MIddle Atmospheric WAter vapor RAdiometer for Campaigns) is a ground-based MWR measuring the pressure-broadened rotational emission line of water vapor at the frequency of 22 GHz. It was also built by the University of Bern and located at Ny-Ålesund, Svalbard (79° N, 12° E). The MIAWARA-C front end is an uncooled heterodyne receiver with a system noise temperature of 150 K. The antenna is followed by a dual polarization receiver. The incident radiation is split into vertical and horizontal polarisation by an orthomode transducer (OMT) located immediately after the feedhorn. The two polarised signals are processed in the two identical receiver chains and separately analyzed in a fast Fourier transform (FFT) spectrometer model Acqiris AC240. The spectrometer has a 400 MHz bandwidth and a spectral resolution of 30.5 kHz. The standard measurement cycle of MIAWARA-C is to measure sky East, reference East, sky West, and reference West for about 15 s each. Every 15 minutes the ambient load is measured for about 2 s and the sky at 60° elevation is measured for about 15 s. A tipping curve is performed to determine the sky temperature at 60° elevation. The difference spectra in the east and west directions and the two polarizations are then calibrated separately with the hot and cold measurements close in time. Similar to GROMOS-C, MIAWARA-C retrieval is also performed with ARTS2 (Eriksson et al., 2011) and QPACK software (Eriksson et al., 2005) according to the optimal estimation algorithm (Rodgers, 2000). An a priori is taken from an MLS climatology of the years 2004-2008. From the measured spectra we retrieve water vapor profiles that cover an altitude range extending from 37 km to 75 km with a vertical resolution of 12-19 km and with a time resolution of 2-4 hours depending on the opacity of the troposphere. For MIAWARA-C retrievals with a constant time resolution and with a constant noise of 0.014K are performed. The AVKs of MIAWARA-C together with its measurement response, errors, and water vapor profiles are shown in Appendix A (Fig. A2). In the upper stratosphere, the errors are 0.5 ppmv and increase from 0.5 ppmv to 1.5 ppmv in the mesosphere. A detailed design of the instrument and description of the retrieval algorithm can be found in Straub et al. (2010) and Tschanz et al. (2013).

## 2.3 Aura-MLS

The Microwave Limb Sounder (MLS) is one of the payloads onboard NASA's Earth Observing System (EOS)-Aura satellite which was launched in 2004 (Waters et al., 2006). The satellite is in a Sun-synchronous orbital altitude of 705 km, with a period of 1.7 hours, and 98 ° inclination. MLS scans the atmospheric limb in the direction of orbital motion which gives almost pole-to-pole coverage (82°S to 82°N), leading to retrieved profiles at the same latitude every orbit, with a spacing of 1.5° great circle angle along the suborbital track. Ozone is retrieved from the band of 240 GHz and water vapor from the 183 GHz line. Temperature is derived from radiances measured from the 118 GHz and 240 GHz channels with a vertical resolution between 3 and 6 km (Schwartz et al., 2015b). The estimated temperature single-profile precision is 0.5 K-1.2 K from 100 to 0.001 hPa. MLS provides ozone profiles (version 5) from 12 to 80 km altitude with a vertical resolution of 2.5-6 km (Schwartz et al., 2015a) and water vapor profiles (version 5) from 10 to 90 km with a vertical resolution of 3.5 km from 316 to 4.64 hPa and 15 km above 0.1 hPa (Lambert et al., 2015). The estimated ozone single-profile precision varies from 0.2 to 0.4 ppmv from the mid-stratosphere to the lower mesosphere. For water vapor, the estimated precision is 0.2–0.3 ppmv in most of the stratosphere and increases to 0.7-0.8 ppmv in the middle mesosphere. It passes at Ny-Ålesund twice a day at around 04:00 and 10:00 UTC. Profiles for comparison are extracted if the location is within $\pm 1.2°$ latitude and $\pm 6°$ longitude of either Ny-Ålesund or the defined virtual conjugate latitude station.

## 2.4 MERRA-2

The Modern-Era Retrospective Analysis for Research and Applications, version 2 (Waters et al., 2006; Gelaro et al., 2017, MERRA-2) is the latest global atmospheric reanalysis produced by the NASA Global Modeling and Assimilation Office (GMAO) from 1980 to the present. MERRA-2 assimilates observation types not available to its predecessor, MERRA, and includes updates to the Goddard Earth Observing System (GEOS) model and analysis scheme so as to provide a viable ongoing climate analysis beyond MERRA's terminus. MERRA-2 provides a regularly-gridded, homogeneous record of the global atmosphere, and incorporates additional aspects of the climate system including several improvements to the trace gas constituents and land surface representation, and cryospheric processes.

In MERRA-2, methods of analysis, model uncertainties, and observations cause uncertainties (Rienecker et al., 2011). Davis et al. (2017) provides a comprehensive assessment of MERRA-2 ozone product that relatively well shows the vertical distribution of ozone and water vapor in the stratosphere and has the best agreement with stratospheric ozone observations compared to other reanalysis products. Wargan et al. (2017) identified that ozone in MERRA-2 data was expected to have higher uncertainties in regions of high variabilities, such as winter high latitudes. The uncertainties are related to the chemistry model ozone bias (Gelaro et al., 2017) which is altitude dependent in MERRA-2. Similar behavior was also found in other GEOS chemistry models (Knowland et al., 2022; Wargan et al., 2023).MERRA-2 stratospheric water vapor is also biased compared to independent observations from Aura-MLS and the Atmospheric Chemistry Experiment-Fourier Transform Spectrometer (ACE-FTS). In this study, we use the ozone and water vapor with 72 model levels from the surface up to 0.01 hPa and a horizontal resolution of $0.5° \times 0.625°$. The time resolution is 6 hours. MERRA-2 products are accessible online through the NASA Goddard Earth

Sciences Data Information Services Center (GES DISC).

## 3  Climatologies of ozone and water vapor

This section examines the ozone and water vapor climatologies over Ny-Ålesund, Svalbard (79° N, 12° E) and conjugate latitude station (79° S, 12° E) generated from GROMOS-C, MIAWARA-C, MERRA-2, and Aura-MLS datasets. It is important to evaluate how well GROMOS-C and MIAWARA-C can monitor the ozone and water vapor variability and enhance our understanding of their distribution in the Arctic middle atmosphere. The inherent variability of ozone and water vapor in the middle atmosphere can be displayed better in the resulting climatologies, as measured by the two instruments involved in this study, and provide a crucial source of data for future work including intercomparison studies and model evaluation, assessing ozone depletion, and validating satellite observations, and studying climate change.

### 3.1  Ozone

Figure. 2 shows the climatological ozone distribution as a function of pressure and time deduced from GROMOS-C, MERRA-2, and Aura-MLS in both hemispheres. Many features of ozone MWR measurement climatologies are broadly consistent with model and satellite data. Some exceptionally maximum values larger than 8 ppmv in MERRA-2 data above 0.1 hPa are described in section 3.

Overall, the ozone profile reveals a characteristic seasonal dependence at polar latitudes. In particular, the altitude of the maximum ozone VMR, as well as its temporal variability, exhibits a seasonality. The peak ozone VMR (approximately 6.5 ppmv) appears in the NH in the late spring, whereas autumn shows the lowest values throughout the course of the year. The maximum observed and reanalysis ozone VMR (approximately 5.5 ppmv) in the SH is 1.0 ppmv smaller than the NH maximum and occurs later in the hemispheric spring season. The primary driver of the hemispheric maximum in ozone VMR is related to circulation processes throughout the stratosphere, including those associated with the BDC, transporting ozone-rich air toward the poles in the winter-spring hemisphere. Precisely, this circulation moves the ozone-rich air from the tropical photochemical source region to high latitude after the polar vortex broke down and essentially enables the intrusion of ozone rich-air from the mid-latitudes into the polar region and replacement of the ozone-depleted air masses. Another essential feature is that GROMOS-C and Aura-MLS capture the tertiary ozone VMR maximum at the northern polar latitude in the early winter and in late spring at the southern polar latitude. Although, the tertiary ozone maximum in GROMOS-C occurs at altitudes close to or even above the limit of 0.8 for the measurement response. MERRA-2 is poor to capture the tertiary ozone VMR maximum in both polar latitudes. Due to the complexity of altered dynamics in the polar regions (Wargan et al., 2017) introducing extra uncertainties into numerical models and data assimilation systems, ozone VMRs exhibit dramatic variability (red shading in Fig. 2b,e) in the mesosphere.

From September to November in the Southern Hemisphere, both MERRA-2 and MLS effectively capture the presence of the ozone hole in the lower stratosphere. However, the climatology of ozone in the NH does not reflect a corresponding signature.

The annual cycle of ozone at 46 hPa in both hemispheres further reveals this significant feature as shown in Fig. 3a. MLS exhibits a greater magnitude of ozone depletion compared to MERRA-2. In Figure 8b, the ozone levels at 3 hPa in the NH follow an annual cycle, with a peak occurring in March and a minimum in October. The SH ozone VMR maintains a constant level of 5 ppmv throughout the summer while experiencing two minimum values in February and October. Furthermore, the ozone VMR in MERRA-2 consistently remains lower than that in MLS, except during autumn in the SH. In the mesosphere, there is relatively good agreement between GROMOS-C and MLS, while MERRA-2 exhibits higher variability (Fig. 3c). Both hemispheres have dramatically different seasonal variations and distribution in ozone due to differences in the stratospheric dynamics of the two hemispheres.

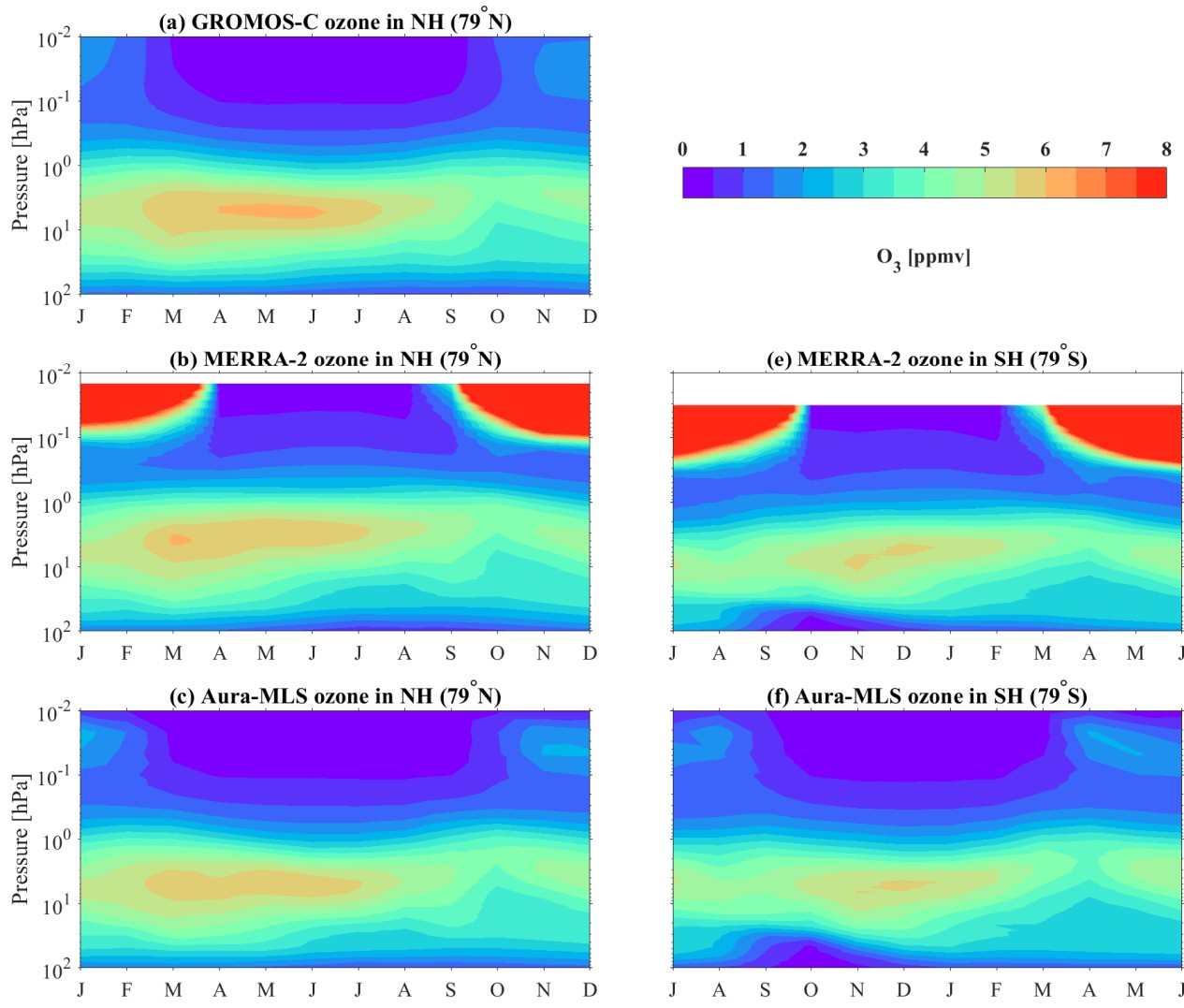

**Figure 2.** Climatology (2015-2021) of the monthly ozone distribution from GROMOS-C (a), MERRA-2 (b, e), and Aura-MLS(c, f) above Ny-Ålesund, Svalbard in the NH and conjugate latitude station in the SH. There is no GROMOS-C measurement for ozone in the SH. The x-axis is the month.

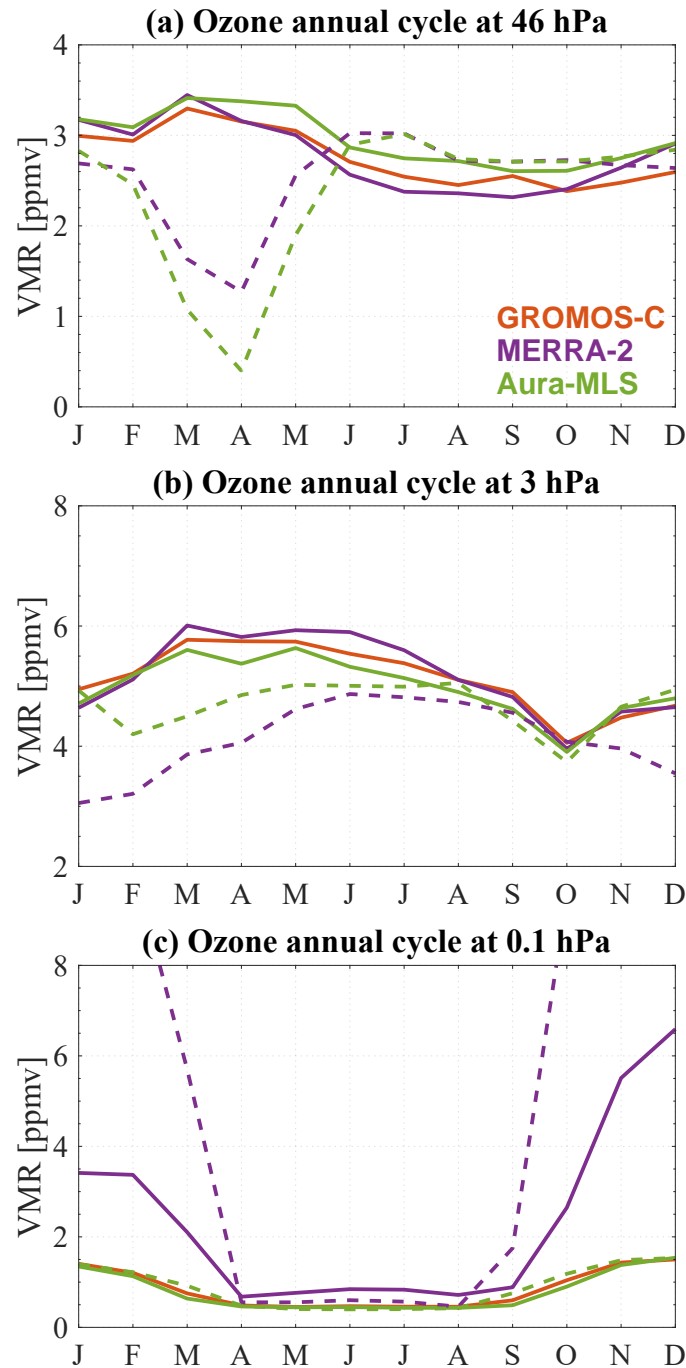

**Figure 3.** The annual cycles of monthly ozone VMR from GROMOS-C, MERRA-2, and Aura-MLS at 46 hPa (a), 3 hPa (b), and 0.1 hPa (c). Solid lines represent Ny-Ålesund in NH and dashed lines represent conjugate latitude station in SH. Each month is averaged for the years 2015-2021. The month on the axes is adjusted accordingly for the SH.

## 3.2 Water vapor

Fig. 4 shows the climatology of water vapor vertical and temporal distribution from MIAWARA-C, MERRA-2, and Aura-MLS over both stations (79° S and 79° N) compiled from measurements collected between 2015 and 2021. The annual variation of water vapor with a maximum during hemispheric summer and a minimum during hemispheric winter is clearly visible. During
220    the hemispheric winter, the middle atmospheric water vapor maximum is shifted down to about 10 hPa while it rises up to the lower mesosphere during the hemispheric summer. The characteristics of water vapor in both hemispheres depend strongly on the mesospheric pole-to-pole circulation which is an upwelling with moist air transporting upwards for the hemispheric summer and a corresponding downward motion with dry air into the stratosphere during the winter (Orsolini et al., 2010). In addition, due to the relatively long photo-chemical lifetime of water vapor, more water vapor produced by methane oxidation
225    accumulates in summer.

The annual cycle of monthly mean water vapor VMR at three separate pressure levels (3, 0.3, and 0.02 hPa) in both hemispheres can be seen in Fig. 5. Water vapor VMR appears at a maximum in October in both hemispheres in the stratosphere (3 hPa) due to the oxidation of methane. Due to photodissociation caused by Solar Lyman-$\alpha$ radiation acting as a sink and the amount of water vapor being in equilibrium between different photochemical processes and vertical transport, water vapor is
230    more smooth with nearly constant mixing ratios from winter to spring in the NH. In the mesosphere (0.3 and 0.02 hPa), water vapor gradient variations can be found during hemispheric spring and summer. The gradient at 0.02 hPa is steeper than at 0.3 hPa from April to July in NH (from November to January in SH). Furthermore, the positive gradient is weaker but the time of increase lasts longer and shows an extreme negative gradient in the hemispheric autumn. In both hemispheres, the seasonal behavior of water vapor is almost symmetric at 0.02 hPa and has a slight asymmetry at 0.3 hPa. At 0.02 hPa the maximum
235    value of the water vapor mixing ratio persists for one month long and at 0.3 hPa the decrease in water vapor is not visible until September in the NH (March in the SH), however then with a steep gradient.

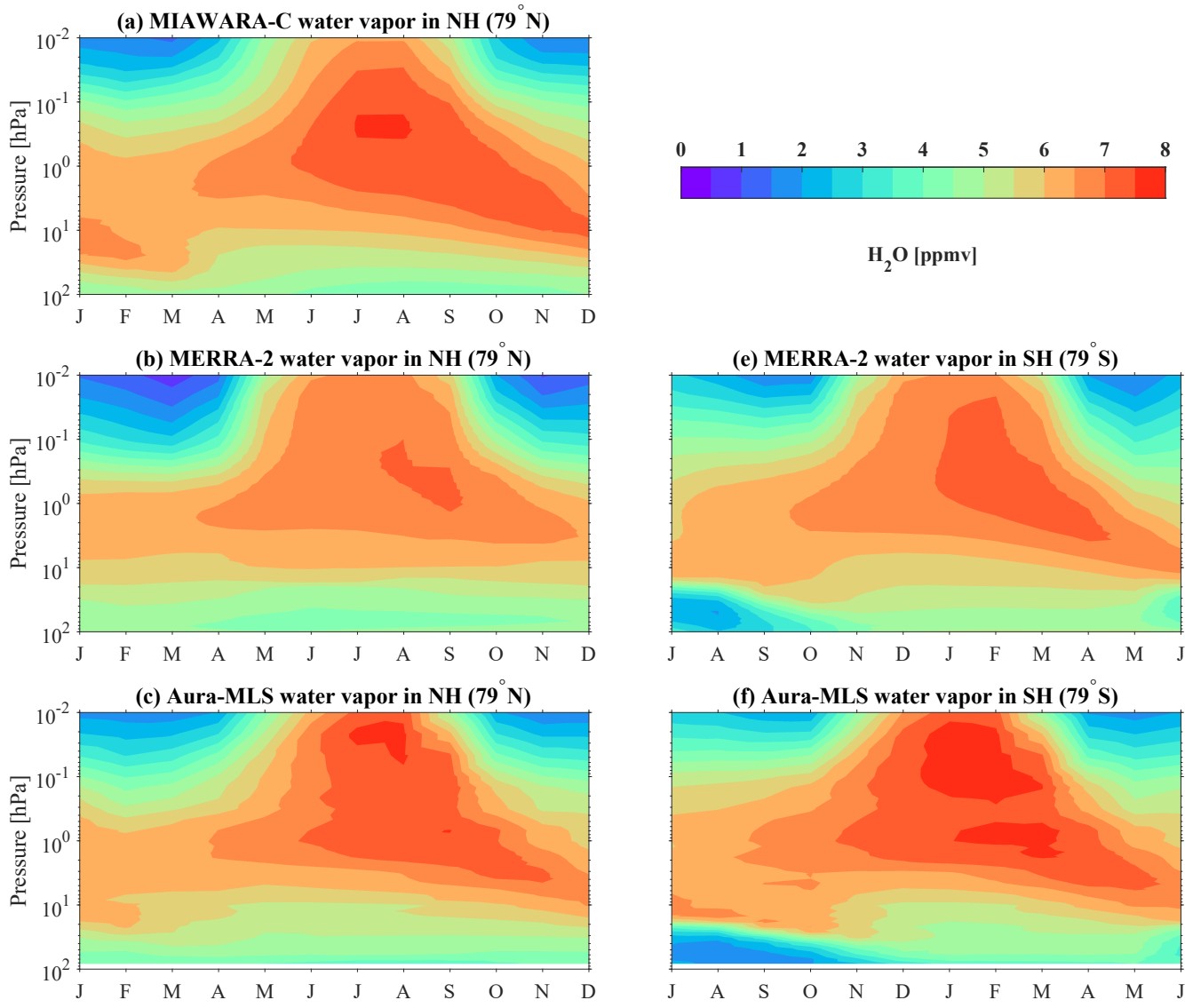

**Figure 4.** Climatology (2015-2021) of the water vapor monthly distribution from MIAWARA-C (a), MERRA-2 (b, e), and Aura-MLS(c, f) above Ny-Ålesund, Svalbard in NH and conjugate latitude station in SH. There is no MIAWARA-C measurement for water vapor in SH.

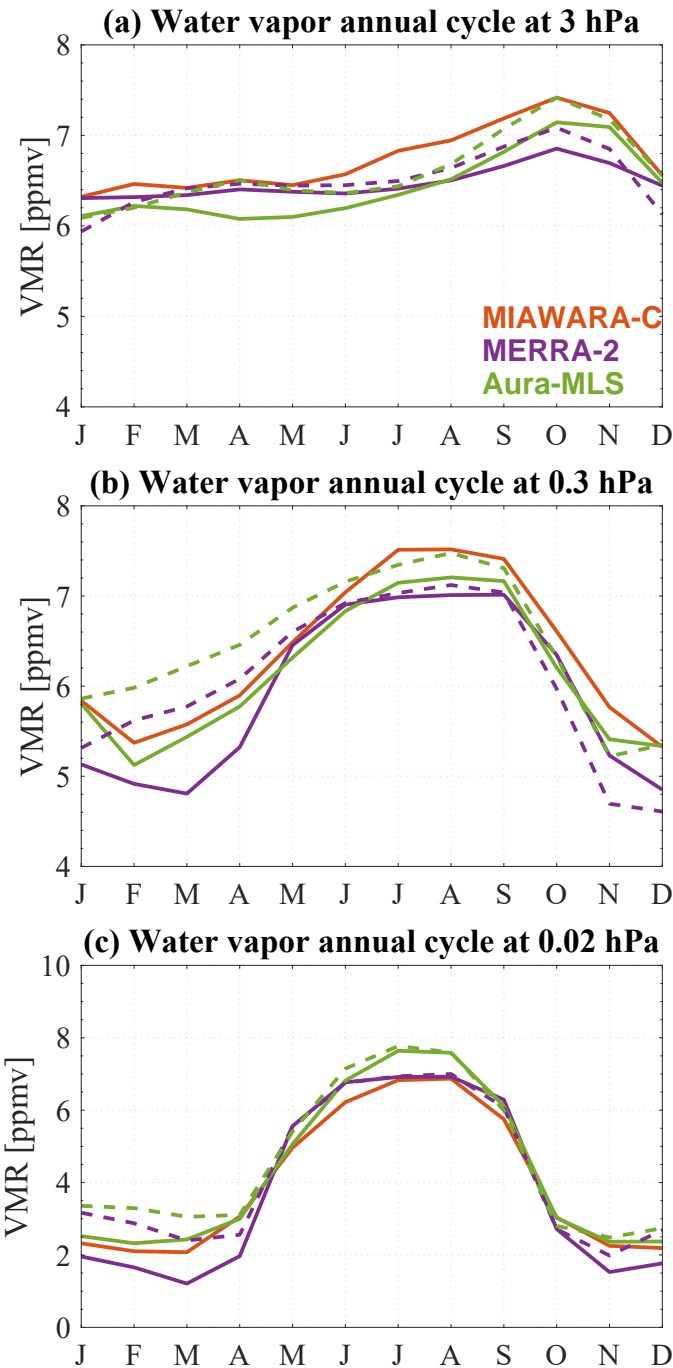

**Figure 5.** The annual cycles of monthly ozone VMR from GROMOS-C, MERRA-2, and Aura-MLS at 3 hPa (a), 0.3 hPa (b), and 0.02 hPa (c). Solid lines represent Ny-Ålesund in NH and dashed lines represent conjugate latitude station in SH. Each month is averaged for the years 2015-2021. The month on the axes is adjusted accordingly for the SH.

### 3.3 Relative differences

In the two ground-based radiometer measurements from GROMOS-C and MIAWARA-C, the aforementioned seasonal be-
havior in their ozone and water vapor distributions display very similar patterns to Aura-MLS and MERRA-2. Still, there are
distinct differences between the datasets. To quantitatively assess the consistency of the ozone and water vapor climatologies
from GROMOS-C and MIAWARA-C, the relative differences (RDs) of ozone and water vapor VMR between the ground-based
MWRs and the two other data sets are calculated using the expression:

$$RD = \frac{(\varphi)_{radiometer} - (\varphi)_{dataset}}{(\varphi)_{dataset}} \cdot 100\% \tag{1}$$

Where $\varphi$ represents either ozone or water vapor VMR. Before evaluating the differences in climatology between MWRs and
MERRA-2 and Aura-MLS, each MERRA-2 and Aura-MLS profile is convolved with the averaging kernels of MWRs. The
convolution is performed according to:

$$\boldsymbol{x}_{conv} = \boldsymbol{x}_a - A(\boldsymbol{x}_a - \boldsymbol{x}) \tag{2}$$

where $\boldsymbol{x}_{conv}$ is the convolved profile, $\boldsymbol{x}_a$ is the a priori profile, A is the averaging kernel matrix and x is the high-resolution
profile of MERRA-2 and Aura-MLS.

Fig. 6 shows the relative differences of ozone and water vapor climatologies from GROMOS-C and MIAWARA-C, respec-
tively, with respect to average convolved MERRA-2 and Aura-MLS. In Fig. 6a, the largest negative RD is larger than 50%
above 0.2 hPa in winter and spring because of the high bias of model ozone chemistry in mesosphere in MERRA-2 (Wargan
et al., 2023). GROMOS-C shows relatively good agreement with MERRA-2 in the lower and middle stratosphere (50-5 hPa),
with RDs smaller than $\pm$5%, but includes a low RD with magnitudes greater than 10% in the upper stratosphere in autumn.
GROMOS-C and Aura-MLS agree well with RDs mainly within $\pm$7% throughout the middle and upper stratosphere and ex-
ceed positive RD 10% in the lower mesosphere (1-0.1 hPa) in March and October (Fig. 6b).

The RD between MIAWARA-C and MERRA-2 is larger than 50% throughout the late autumn to spring months above 0.2 hPa
in Fig. 6c. Simultaneously, MERRA-2 underestimates water vapor VMR at all altitudes due to the lack of assimilated obser-
vation to constrain the water vapor reanalyses in the polar mesosphere and in part to the methane oxidation parameterization
being disabled in the GEOS Composition Forecast model (Davis et al., 2017; Knowland et al., 2022). The MIAWARA-C agrees
with MERRA-2 within 7% in the stratosphere and lower mesosphere (about 100–0.3 hPa). MIAWARA-C and Aura-MLS show
relatively good agreement, RDs within 5% throughout the stratosphere and lower mesosphere (approximately 100–0.02 hPa)
in Fig. 6d. Furthermore, MIAWARA-C exhibits a negative RD within 8% above the lower mesosphere (around 0.1-0.01 hPa).
Additionally, we found a wet bias of 5-7% between 100 and 1 hPa for MIAWARA-C measurements in each season. The wet
bias becomes noticeable between July and November, primarily attributed to variations in instrument performance and, to a
certain extent, the dynamic effects causing an elevation in water vapor levels within the stratosphere (as seen in Fig. 8a).

Note that the RD between ground-based MWRs and Aura-MLS is in part the Aura-MLS ozone and water vapor profile sam-
pling (as mentioned in Sect. 2.3) and the measurement geometry, leading to seasonal variations in the polar ozone and water
vapor distribution. Furthermore, the diurnal cycle in the ozone and water vapor has not been explicitly accounted for in the

GROMOS-C and MIAWARA-C measurements. Neglecting the diurnal cycle potentially contributes to positive RDs between MWR measurement and other data sets in the upper stratosphere and lower mesosphere. Overall, GROMOS-C and MIAWARA-C are valuable to monitor the distribution of stratospheric ozone and mesospheric water vapor at the polar latitudes, respectively, which gives us more details to investigate their long-term variability, sources, and trend.

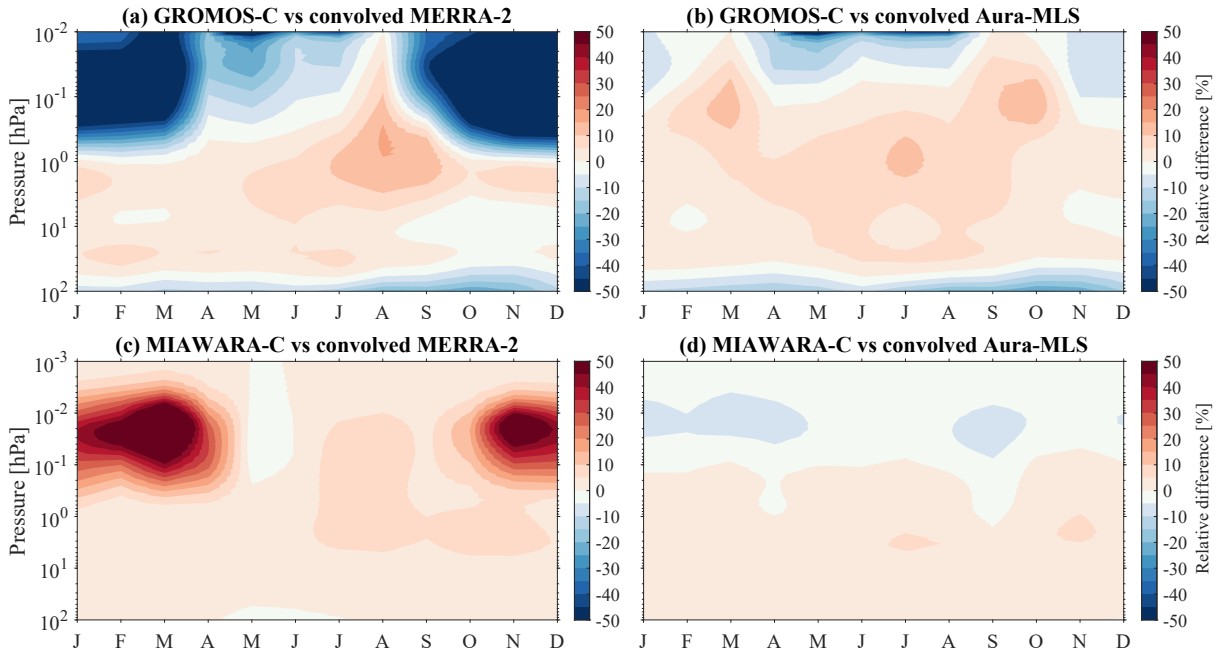

**Figure 6.** Monthly distributions of ozone and water vapor relative differences between ground-based MWRs and data sets: (a, c) MERRA-2, (b, d) Aura-MLS.

## 4   Time series of ozone and water vapor

### 4.1   Ny-Ålesund, Svalbard (79° N, 12° E) in the NH

The time series of daily ozone for GROMOS-C, MERRA-2, and Aura-MLS at Ny-Ålesund, Svalbard (79° N, 12° E) extending from 2015 to 2021 are shown in Fig. 7. The ozone daily profiles measured with GROMOS-C cover a pressure range of 100-0.03 hPa which corresponds to about 16-70 km. The horizontal upper and lower white lines indicate the bounds of the trustworthy pressure range where the measurement response is larger than 0.8, meaning that the measured spectrum contributes more than 80% to the retrieved profile. The measurement data gaps are that GROMOS-C measured CO for about 2 months during winter 2017/2018 and the spectrometer had a hardware problem during winter 2016/2017 and summer 2019.

Fig. 7 reveals the annual ozone cycle with higher ozone VMR in summer (about 6 ppmv) than in winter (about 4.5 ppmv) at about 5 hPa (≈35 km). GROMOS-C ozone VMR time series together with MERRA-2 and Aura-MLS at about 5 hPa smoothed by a 30-day running mean is shown in Appendix B (Fig. B1a). We can see that all three datasets are able to capture the annual

ozone variations well in the stratosphere. Stratospheric ozone largely follows the annual cycle of solar irradiation and is produced through the Chapman cycle, in particular, ozone VMR is mainly dominated by photochemical production in the summer months in the Arctic middle atmosphere.

In late winter and spring, the stratospheric ozone's higher variability is largely associated with the stratospheric polar vortex. Fig. 7 presents the ozone VMR starts to increase up to the maximum value of about 8 ppmv for some of the years in the stratosphere when the polar vortex is disturbed or weakened by the planetary waves leading to the formation of SSWs. It shows that the planetary wave activity results in meridional transport of the ozone-rich air from the subtropics towards the pole, and perturbed significantly the distribution of ozone. For example, the polar vortex split and shifted away from Ny-Ålesund, ozone VMR reached about 7 ppmv during the winter 2018/2019 SSW (Schranz et al., 2020). In some years, the polar vortex is stable and strong over Ny-Ålesund, and ozone VMR sustains smaller values. At the end of the winter 2019/2020 season, the stratosphere featured an extremely strong and cold polar vortex resulting in low stratospheric ozone in the polar regions (Lawrence et al., 2020b; Inness et al., 2020). During late winter and early spring, stratospheric ozone decreases rapidly when the vortex passes over Ny-Ålesund.

Marsh et al. (2001) and Smith et al. (2009, 2018) investigate the tertiary ozone maximum in the winter middle mesosphere at high latitudes based on the models and observations. In this study, GROMOS-C and Aura-MLS present the seasonal tertiary ozone layer at 0.03–0.02 hPa (about 70–75 km) in winter months, but the tertiary ozone VMRs have higher values of 15%–20% in MERRA-2 (as shown the red shading in Fig. 7b). The red shading originates the uncertainties in the MERRA-2 product which are expected to be magnified at high latitudes in winter and spring when the variability is increased compared to other seasons (Wargan et al., 2017). The anomalous atmospheric dynamics, displaced/split polar vortex, and hemispherically asymmetric conditions during SSW may cause complexity and additional uncertainties in the estimation of ozone flux/transport terms. Fig. 7 shows the consistencies of GROMOS-C with both MERRA-2 and Aura-MLS datasets in the time series of ozone below 1 hPa. A clear annual cycle in the stratosphere is well captured by all datasets, and the higher variability of ozone in winter and spring seasons is clearly visible. Remarkably, the annual variation in MERRA-2 ozone is significantly different from the observed variation throughout the mesosphere above 1 hPa.

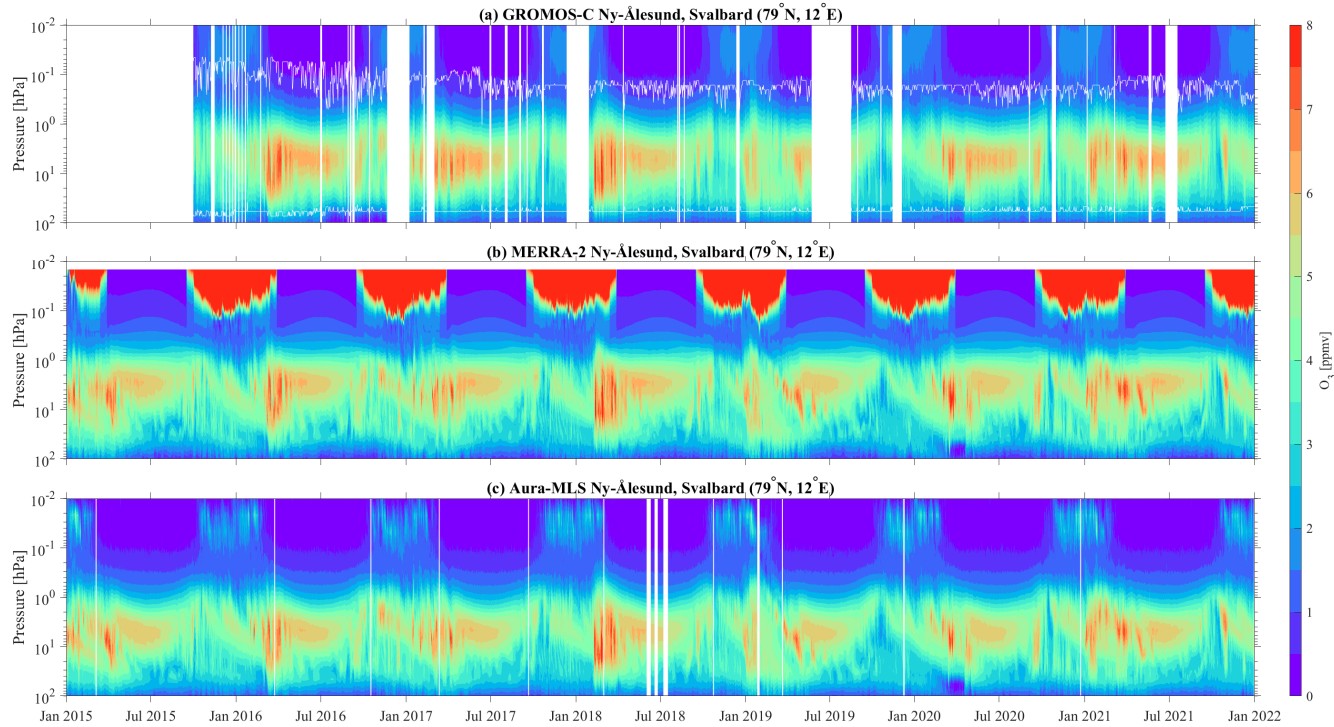

**Figure 7.** Time series of daily ozone VMR as a function of pressure over Ny-Ålesund, Svalbard (79° N, 12° E) for the 2015-2021 period. Panels show ozone VMR from (a) GROMOS-C measurements, (b) MERRA-2 reanalysis data, and (c) MLS satellite observations. The vertical white lines represent the data gaps caused by the hardware and measurement problems. The horizontal upper and lower white lines indicate a measurement response of 0.8 in Fig. 7a.

Fig. 8 shows the time series of daily water vapor VMR from MIAWARA-C, MERRA-2, and Aura-MLS at Ny-Ålesund, Svalbard (79° N, 12° E) for the period 2015-2021. MIAWARA-C measures continuously water vapor profiles which cover a pressure range from 5-0.02 hPa corresponding to about 37-75 km. The horizontal upper and lower white lines indicate again the bounds of the trustworthy pressure range where the measurement response is larger than 0.8.

The most evident feature of water vapor is its annual cycle with higher mixing ratios during local summertime and lower mixing ratios during local wintertime throughout the middle atmosphere. This seasonal behavior is mainly driven by the upward and downward branches of the mesospheric residual circulation. As shown in Appendix B (Fig. B1a), water vapor VMR has a maximum of about 7.5 ppmv in summer and a minimum of about 3.5 ppmv in winter at 0.1 hPa (approximately 60 km). Due to the air subsidence inside the polar vortex from autumn to winter, water vapor VMR reaches the maximum at 10 hPa.

In some years such as 2020, water vapor exhibits a larger variability in late winter and spring. The variation of water vapor is mainly affected by the occurrence of a major SSW which interrupts the polar vortex, after that, the vortex recovers, and the lower mesospheric water vapor content increases and is accompanied by a decrease in the stratosphere, corresponding to the water vapor vertical profile in the regions outside of the polar vortex (Schranz et al., 2019, 2020). In general, MIAWARA-C the

annually varying mesospheric distribution of water vapor agrees well with reanalysis data and satellite observations. However, there are notable differences at the polar latitudes in the water vapor VMR compared to MERRA-2 throughout the stratosphere and the mesosphere. While MERRA-2 shows a tendency to lower water vapor VMR compared to MIAWARA-C and MLS observations, the seasonal variations of similar amplitude do show reasonably good agreement with the observations. This tendency is likely related to the lack of assimilated observations and known deficiencies in the representation of stratospheric transport (Davis et al., 2017) in the reanalysis data.

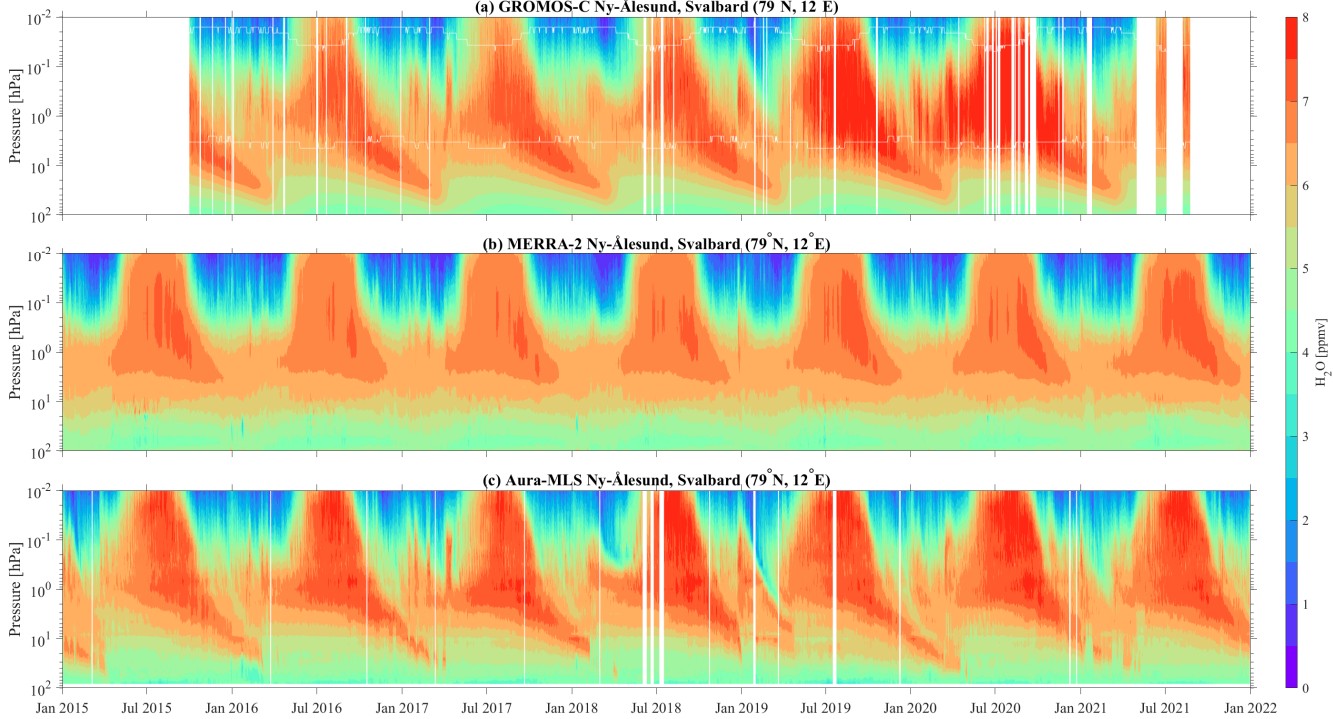

**Figure 8.** Time series of water vapor VMR as a function of pressure over Ny-Ålesund, Svalbard (79° N, 12° E) for the 2015-2021 period. Panels show ozone VMR from (a) MIAWARA-C measurements, (b) MERRA-2 reanalysis, and (c) MLS satellite observations. The vertical white lines represent the data gaps caused by the hardware and measurement problems. The horizontal upper and lower white lines indicate a measurement response of 0.8 in Fig. 8a.

## 4.2 Conjugate latitude station (79° S, 12° E) in the SH

Fig. 9 and Fig. 10 show time series of ozone and water vapor VMR from MERRA-2 and Aura-MLS at conjugate latitude station (79° S, 12° E) for the 2015-2021 period, respectively. For comparison purposes, the conjugate latitude station results are lagged by 6 months relative to those for Ny-Ålesund. Both stations exhibit annual cycles of ozone while the conjugate latitude station shows an ozone VMR maximum of about 6 ppmv in summer and a minimum of about 4 ppmv in winter at

about 5 hPa in Appendix B (Fig. B1b).

Compared to the interannual ozone variability at Ny-Ålesund, the results from the conjugate latitude station are less variable throughout the spring in the stratosphere. Compared to the NH, the planetary wave activity is much weaker in the SH where a minimum in the ozone mixing ratios prevails over the polar latitudes during the late winter and spring. There is a dominance of

an isolated and stable polar vortex which inhibits the meridional ozone transport to the south pole throughout the annual cycle, and the formation of polar stratospheric clouds promotes the production of chemically active chlorine and bromine leading to catalytic ozone depletion in the lower stratosphere. This process is well-reflected in the MLS observations and MERRA-2 data for September/October at the conjugate latitude station. Fig. 10 presents the water vapor annual cycle with higher mixing ratios during local wintertime and lower mixing ratios during local summertime over the conjugate latitude station.

Furthermore, polar stratospheric cloud particles can sediment with considerable velocities and irreversibly remove water and nitric acid resulting in a substantial reduction of water vapor VMR at the lower stratosphere during the southern polar winter and early spring (Waibel et al., 1999; Tritscher et al., 2021). After September, the water vapor VMR increases again as the polar stratospheric cloud influence reduces.

While MERRA-2 generally does an excellent job in reproducing the variability of stratospheric ozone and water vapor, it

fails in the mesosphere as discussed in previous sections, and also underestimates the vertical extent of the ozone hole, which appears to end at lower altitudes (larger pressures) in MERRA-2 than in MLS. This is also reflected in water vapor, where MERRA-2 fails to reproduce the vertical layering (e.g. July 2015 or July 2017), and underestimates the area of dehydration. Since the reanalysis is less constrained when it is assimilated (Davis et al., 2017; Shangguan et al., 2019), this behavior can be seen in MERRA-2 compared to observations. It further demonstrates that ozone and water vapor in the NH will continue to be

available from ground-based MWR observations, however, the detailed information about the SH winter as shown here for the 'conjugate latitude' will be lost after the end of the last three limb sounders such as Aura-MLS which is still observing.

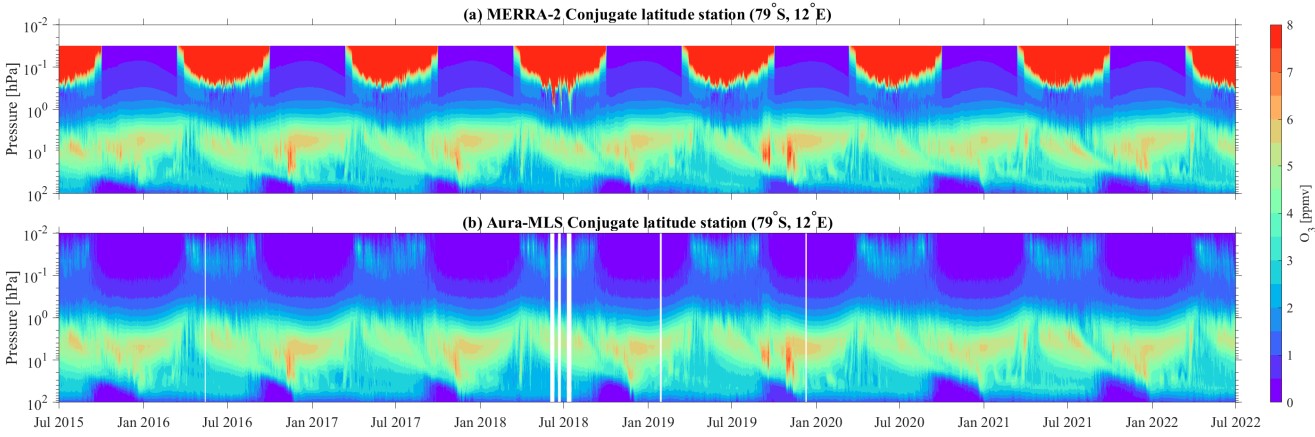

**Figure 9.** Same as Fig. 7 (without GROMOS-C measurements) but for the SH at conjugate latitude station (79° S, 12° E).

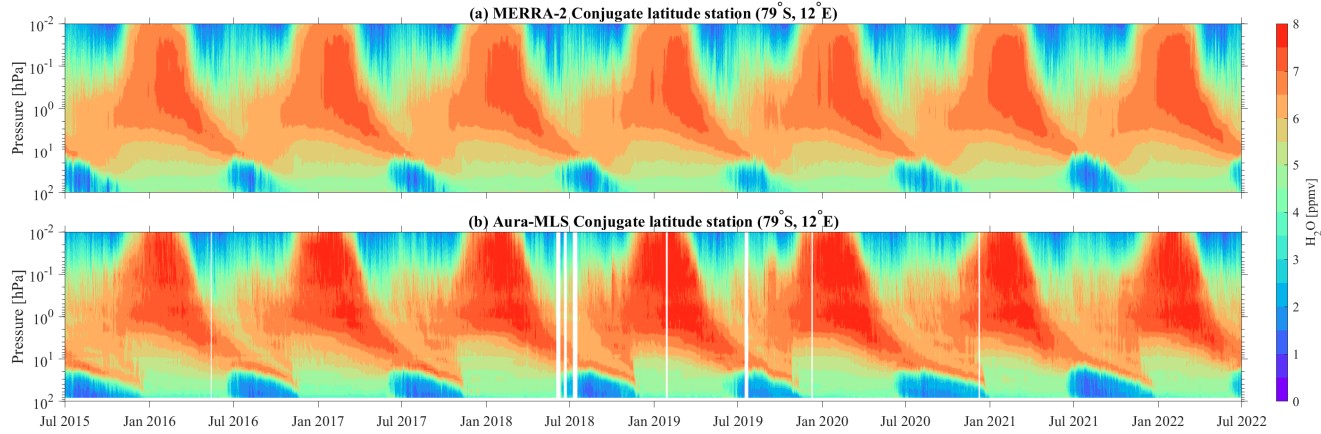

**Figure 10.** Same as Fig. 8 (without MIAWARA-C measurements) but for the SH at conjugate latitude station (79° S, 12° E).

## 5   Dynamics and transport of water vapor

Water vapor is widely used as a tracer to investigate the dynamics of transport processes in the Arctic middle atmosphere (Lossow et al., 2009; Straub et al., 2012; Tschanz et al., 2013; Schranz et al., 2019, 2020). The chemical lifetime of water
vapor is of the order of years in the lower stratosphere, months in the lower mesosphere, and weeks in the upper mesosphere (Brasseur and Solomon, 2005). Water vapor mixing ratios display different dynamical features depending on the altitude ranges because of different vertical gradients of water vapor VMR above and below its peak in the upper stratosphere. The water vapor mixing ratio is assumed to be constant for one month and a half, and the vertical velocity of air can be estimated.

The time periods of the year when water vapor VMR increases and decreases with altitude (in the upper stratosphere and
lower mesosphere) measured from MIAWARA-C and Aura-MLS over both Ny-Ålesund and conjugate latitude stations are well given in this study. We will denote the period with a stable high water vapor mixing ratio as the hemispheric summer and the positive and negative transition periods as the upwelling and downwelling branches, respectively. With the northern and southern hemispheric water vapor measurements, we calculate the effective ascent and descent rates as derived from a linear regression fit to different water vapor mixing ratio isopleths (5.5, 6.0, and 6.5 ppmv). For instance, the ascent and descent rates
from 6.5 ppmv water vapor isopleth are shown in Fig. 11 and Fig. 12).

The time period of many years of descent rate from 15 September to 31 October in the altitude range of about 50-70 km is well presented in Fig. 11 (the first and third rows). This is an estimated result not a quantitative calculation of the water vapor descent since the water vapor dynamic and chemicals reactions not directly related to descent may also affect the polar water vapor changes. However, the effect of dynamic processes such as planetary wave disturbance is relatively obvious to estimate
the ascent rate. As shown in Fig. 11 (second and fourth rows), the time period of five years (2015, 2016, 2017, 2018, and 2021) of ascent rate starts from 05 May to 20 June and another two years (2019 and 2020) the starting time for the increase of water vapor happens earlier by approximately 20 days which is from 15 April. MIAWARA-C and Aura-MLS observe in 2019 and

2020 years the return of the water vapor mixing ratio to pre-winter values in mid-April and the moist air is lifted to greater altitudes in early May. The starting time of the ascent rate in 2019 and 2020 is about three weeks earlier than that in other years,

380 which is caused by several processes. The planetary waves displace the polar vortex above Ny-Ålesund, and water vapor-rich air is transported into the upper stratosphere, and furthermore, the maximum water vapor mixing ratio at the altitude of about 55 km is re-established. Wave breaking and mixing above the strongest vortex level tends to reduce the tracer gradient (Lee et al., 2011), leading to an increase in the water vapor mixing ratios. In addition, photochemical processes from solar radiation also can contribute to the accumulation of water vapor in the springtime at the stratopause altitude (Brasseur and Solomon,

385 2005).

In the SH, the time period of each year of the ascent rate is relatively consistent from 15 October to 30 November in the upper stratosphere and mesosphere (Fig. 12). The descent rate of water vapor from 15 March to 30 April in the SH appears to be similar for each of the seven years, likely due to the higher stability and strength of the southern polar vortex.

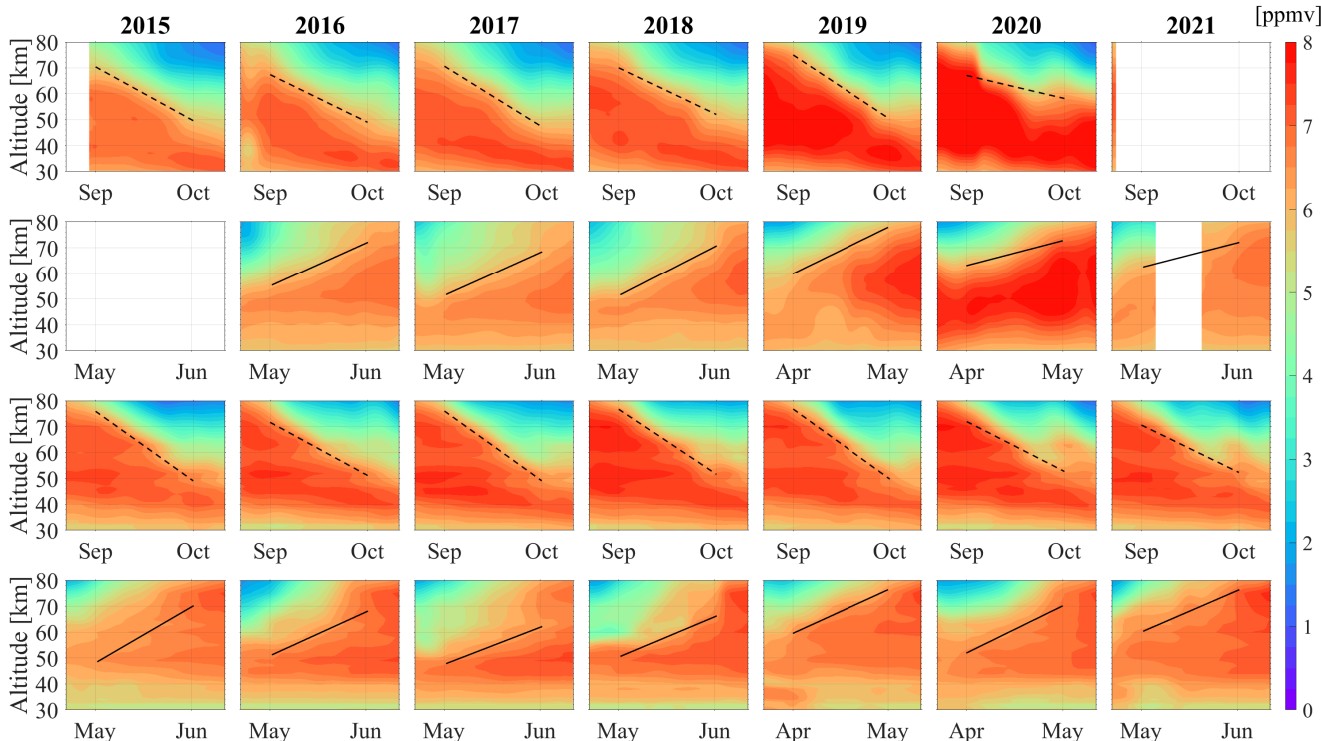

**Figure 11.** Time-altitude of mean water vapor VMR from MIAWARA-C (first and second rows) and Aura-MLS (third and fourth rows) over Ny-Ålesund, Svalbard (79°N, 12°E) in the NH. The data has been smoothed by a 20-day Gaussian. The black dashed and solid lines indicate the descent and ascent rates of water vapor as derived from a linear regression fit to the 6.5 ppmv isopleth of water vapor VMR, respectively. There is no data for MIAWARA-C from May to June 2015 and from September to October 2021.

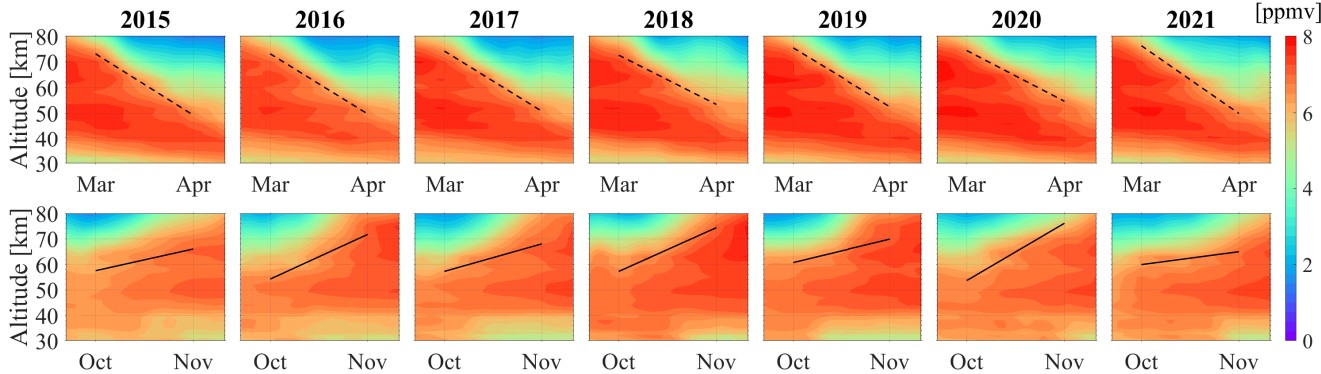

**Figure 12.** Same as Fig. 11 (without GROMOS-C measurements) but for conjugate latitude station (79° S, 12° E) in the SH.

Fig 13 shows the annual variability of effective ascent and descent rates in the southern and northern hemispheres during a given time period. In contrast to the NH, the southern hemispheric rate exhibit less interannual variability and an approximately consistent variation with the rates corresponding to the different isopleths, illustrating the qualitative agreement in different years in the effective vertical rates for the transition periods in the SH. The uncertainties for the ascent and descent rates of airflow over northern and southern polar latitude stations are depicted by error bars in Fig 13. Effective ascent and descent rates estimated from water vapor measurements are very significant. Table 1 gives a more quantitative perspective to compare the vertical movement of air in both hemispheres. In the NH, the average vertical velocities are $3.4\pm1.9$ mm s$^{-1}$ from MIAWARA-C and $4.6\pm1.8$ mm s$^{-1}$ from Aura-MLS for upwelling from spring to summer, and $5.0\pm1.1$ mm s$^{-1}$ from MIAWARA-C and $5.4\pm1.5$ mm s$^{-1}$ from Aura-MLS for downwelling from summer to autumn. During the transition from winter to spring, the vertical velocity is $5.2\pm0.8$ mm s$^{-1}$ for downwelling and $2.6\pm1.4$ mm s$^{-1}$ for upwelling from autumn to winter calculated by Aura-MLS in the SH. Table 1 shows a stronger upwelling branch in the NH polar summer mesosphere as compared to the SH, accompanied by a stronger downwelling branch towards the SH winter in the polar region. In general, these results assess the ability to derive middle atmospheric ascent and descent rates from water vapor measurements at polar latitudes and further provide evidence for the higher variability in the NH than in the SH.

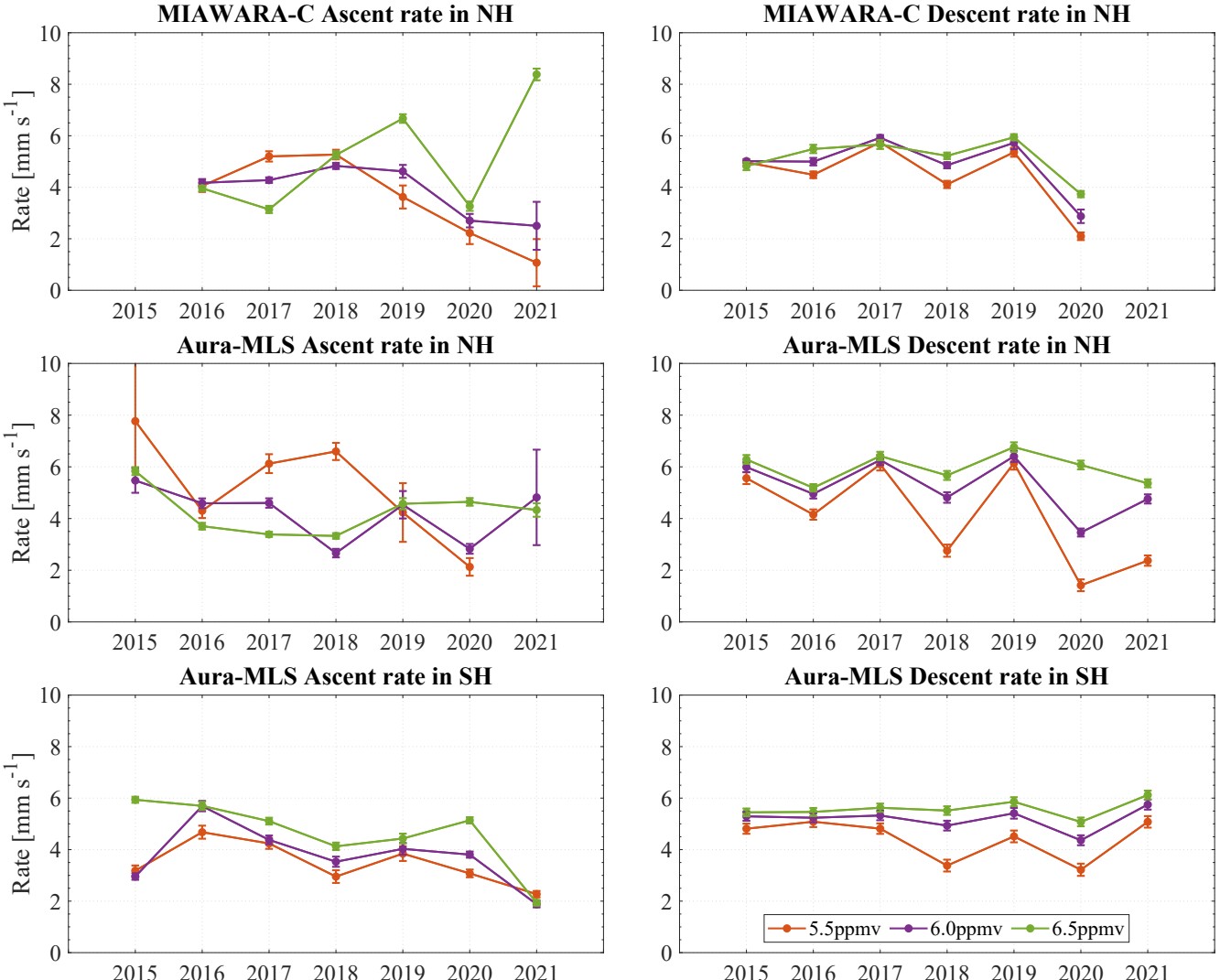

**Figure 13.** Interannual variability of the effective ascent and descent rates over both stations Ny-Ålesund, Svalbard (79°N, 12°E) and conjugate latitude station (79°S, 12°E) estimated from the linear regression fit to the 5.5, 6.0, and 6.5 ppmv isopleths of water vapor VMR from MIAWARA-C and Aura-MLS. Bars indicate the uncertainties in the ascent and descent rates based on the MIAWARA-C and Aura-MLS observations.

## 6  Discussion

Continuous observations of essential climate variables such as ozone and water vapor are important for investigating the ra-
405  diative balance of the atmosphere. Interhemispheric and interannual differences shed light on ozone and water vapor natural
variability due to transport and photochemistry. Ground-based measurements such as those performed by GROMOS-C and

**Table 1.** Comparison of the mean values and standard deviations (SDs) of ascent and descent rates from MIAWARA-C and Aura-MLS in both hemispheres. The mean is computed for the rates from 5.5, 6.0, and 6.5 ppmv isopleths (as shown in Fig 13).

| Ascent rate (mm s$^{-1}$) | NH (mean±SD) | SH (mean±SD) |
|---|---|---|
| MIAWARA-C | 3.4±1.9 | - |
| Aura-MLS | 4.6±1.8 | 2.6±1.4 |

| Descent rate (mm s$^{-1}$) | NH (mean±SD) | SH (mean±SD) |
|---|---|---|
| MIAWARA-C | 5.0±1.1 | - |
| Aura-MLS | 5.4±1.5 | 5.2±0.8 |

MIAWARA-C permit to sustain of a high-resolution and continuous data set of these trace gases at remote locations such as Ny-Ålesund. A comparison to the Aura-MLS satellite data exhibits excellent agreement throughout all altitudes in the stratosphere to GROMOS-C, and the upper stratosphere and lower mesosphere to MIAWARA-C, respectively. The climatologies Aura-MLS and the MWRs agree within ±10% during the year. However, a climatological comparison to the reanalysis data MERRA-2 and our ground-based radiometers indicates larger discrepancies above 0.2 hPa. These increased deviations are partially understandable by the implemented radiative transfer schemes and other model physics such as interactive chemistry, which is computationally much more expensive (Gelaro et al., 2017). Furthermore, MERRA-2 includes the MLS observations of temperature and ozone in the 3DVAR data assimilation (Wargan et al., 2017). MLS observations are most important for the mesosphere and are weighted by their precision and accuracy.

The interhemispheric comparison is performed by defining a virtual conjugate latitude station (79° S, 12° E) in the southern hemisphere at the conjugate geographic coordinates. Although the general seasonal morphology is very similar, the northern hemisphere shows much more variability in the time series of ozone and water vapor. One of the most significant differences is the occurrence of the ozone hole in the southern hemisphere towards the end of the winter season below 10 hPa (Solomon et al., 2014). During this time also the water vapor VMR measurements exhibit a minimum due to the formation of polar stratospheric clouds (Flury et al., 2009; Bazhenov, 2019) providing even more favorable conditions for catalytic ozone destruction reactions. In Fig 14, we present zonally averaged temperature at two conjugate latitudes comparison results between the polar and mid-latitudes from Aura-MLS observations in both hemispheres. The southern polar latitudes appear much colder about 20 K than their northern counterparts as seen in Appendix C (Fig C1). Furthermore, the temperature gradient between the mid- and high-latitudes is stronger in the southern hemisphere at the stratosphere and, thus, drives a more stable polar vortex due to the thermal wind balance, which prevents the mixing of ozone-rich air from the low- and mid-latitude into the polar cap by planetary waves as it often can be observed in the northern hemisphere (Schranz et al., 2019). There are only a few occasions from 2015 where in the northern hemisphere such a stable and cold polar vortex was observed in 2015/2016 and 2019/2020 in the Arctic winter(Matthias et al., 2016; Lawrence et al., 2020a), which also can lead to anomalies in the middle atmospheric

dynamics (Stober et al., 2017).

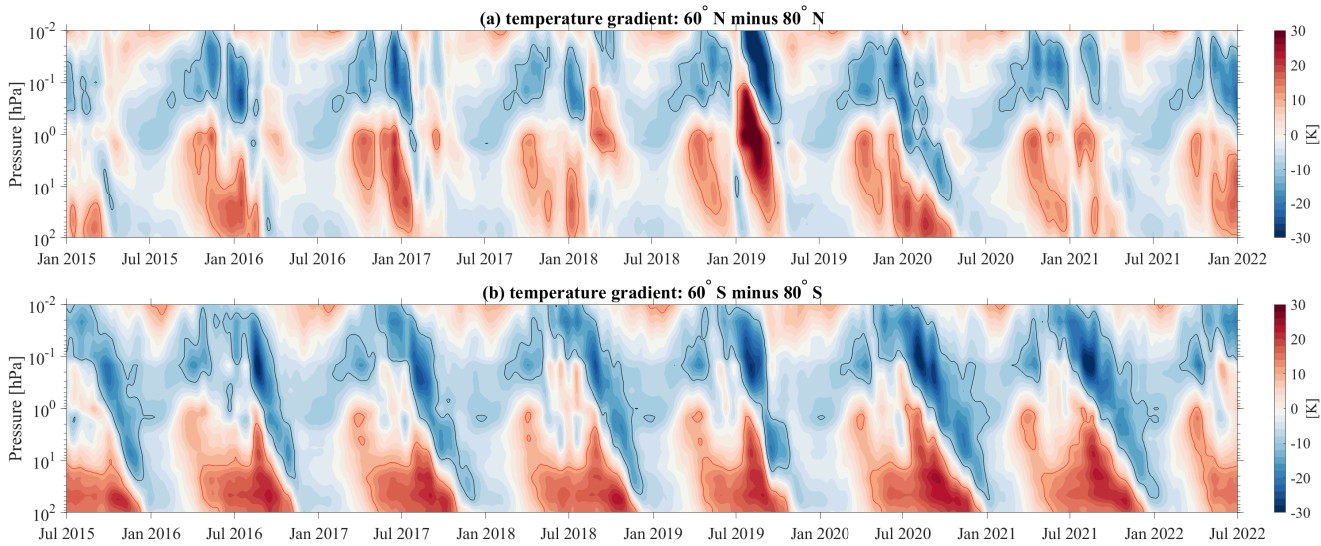

**Figure 14.** Temperature gradients between the mid-latitude and polar latitudes in the NH (60° N & 80° N) and SH (60° S & 80° S), respectively. Red and black contours represent positive and negative values (±10 K).

In addition to the strength of the polar vortex, we investigated the strength of the upwelling and downwelling in both hemispheres, which we consider a proxy of the strength of the residual circulation. Water vapor has a longer lifetime than CO at 50-70 km (Brasseur and Solomon, 2005), which makes it a more robust tracer to manifest the vertical motion of air. The
effective rates of vertical transport are estimated by using the water vapor measurements following the approach presented by Straub et al. (2010). The vertical velocities are derived under the assumption that other processes such as chemical reactions, photodissociation, or horizontal advection are less important. Here we analyze periods around the equinoxes. During this summertime of the year, the horizontal gradients of temperature between the polar region and the mid-latitudes are minimal or negligible and, thus, there is no strong forcing due to horizontal temperature gradients or planetary waves driving the zonal
or meridional transport. Furthermore, we only investigated altitudes below typical OH-layer heights, which also reduces the impact of photodissociation in our estimates (Ryan et al., 2018). We obtained vertical velocities of 3.4±1.9 and 4.6±1.8 mm s$^{-1}$ for the upwelling towards the northern hemispheric summer and vertical motions of 5.0±1.1 and 5.4±1.5 mm s$^{-1}$ downwelling during the fall transition. Furthermore, there is rather significant interannual variability in the northern hemisphere that is not found above the Antarctic continent. Due to the frequent occurrence of SSWs, the spring transition is more variable in the
NH (Matthias et al., 2021). In the southern hemisphere, the spring transition towards summer is characterized by 2.6±1.4 mm s$^{-1}$ vertical velocity, and during the transition from summer to winter, the downwelling with 5.2±0.8 mm s$^{-1}$ vertical velocity is observed. A high variability in NH winter downwelling would imply a high variability of SH summer upwelling, which is indeed observed in Fig 13. This suggests that the interhemispheric coupling (Körnich and Becker, 2010; Orsolini et al., 2010;

Smith et al., 2020a) is that disturbances are transmitted from the winter stratosphere to the summer mesosphere in the other hemisphere. However, it is worth noting that there is a lower variability in SH winter downwelling, which corresponds to a higher variability of NH summer upwelling. This observation suggests that the coupling between the winter and summer circulations within each hemisphere is influenced by various factors beyond the interhemispheric exchange. As shown in Fig 14, with a stronger temperature gradient during SH winter, the polar vortex is relatively stable and well-defined, leading to reduced variability in the downward motion of air (downwelling). For instance, the SSW events during the NH winter (Limpasuvan et al., 2016; Schranz et al., 2019, 2020) or increased stratospheric planetary wave activity (De Wit et al., 2015) lead to increased variability in the NH winter stratosphere. Simulations with a GW resolving model response to the enhanced winter hemisphere Rossby-wave activity may lead to both interhemispheric couplings through a downward shift of the GW-driven branch of the residual circulation and an increased GW activity at high summer latitudes (Becker and Fritts, 2006). Other observational studies using polar mesospheric clouds as well as stratospheric reanalysis data exhibited an interhemispheric correlation during the summer months (Karlsson et al., 2007; Espy et al., 2011). More recent model results suggested that the strongest interhemispheric coupling signatures are found between stratosphere and mesosphere in the opposite hemispheres (Smith et al., 2020b).

## 7 Conclusions

Continuous ground-based measurements of ozone and water vapor remain an essential tool to understand the short and long-term evolution of the middle atmosphere, as well as for the validation and parameterization of atmospheric models. In this study, we present ozone and water vapor measurements from the two ground-based radiometers GROMOS-C and MIAWARA-C located at Ny-Ålesund, Svalbard collected between 2015 and 2021. The data were compared to observations from MLS onboard the Aura spacecraft as well as reanalysis data MERRA-2. This comparison showed a good agreement for the climatological behavior between the ground-based radiometers and MLS to within almost $\pm7\%$ for ozone at about 50–1 hPa, within $\pm5\%$ for water vapor at about 100–0.02 hPa. However, we identified pronounced differences between the measurements and the reanalysis data above 0.2 hPa where MERRA-2 deviations up to 50% were visible. Ground-based observations are going to become more important within the next years as the satellite instruments such as MLS are going to reach the end of their life and so far there are no adequate replacements in orbit.

By defining a virtual conjugate latitude station in the southern hemisphere, we investigated altitude-dependent interhemispheric differences. Both trace gases showed a much higher variability during the northern hemispheric winter driven by planetary wave activity. The southern hemisphere was characterized by a more stable polar vortex and colder temperatures in the polar cap that results in more favorable conditions to form polar stratospheric clouds and, thus, more efficient ozone destruction by catalytic reactions causing the well-known ozone hole. Furthermore, the polar stratospheric cloud formation was accompanied by a reduction of the water vapor VMR at the same altitudes in the lower stratosphere.

We investigated the strength of the residual circulation by estimating the up- and downwelling above Ny-Åesund and the corresponding conjugate latitude station. Typical ascent rates during the summer transition reach values of 3.4-4.6 mm s$^{-1}$

and for the downwelling, in the fall transition, vertical velocities of 5.0-5.4 mm s$^{-1}$ are inferred. Correspondingly the vertical velocity of 2.6 mm s$^{-1}$ for the upwelling and 5.2 mm s$^{-1}$ for the downwelling is calculated in the SH. The northern hemisphere also reflected a much more pronounced interannual variability compared to the southern polar latitudes. However, there is no strong correlation between up- and downwellings in the opposite hemispheres most likely due to dynamic processes such as the QBO or weather patterns which play a role and need to be taken into account. Therefore, long-term ozone and water vapor measurements will improve a deeper understanding of the mechanisms which control polar ozone and water vapor variability and predict the future evolution of middle atmospheric ozone and water vapor in climate changes.

*Data availability.* The GROMOS-C and MIAWARA-C level 2 data are provided by the Network for the Detection of Atmospheric Composition Change and are available at http://www.ndacc.org (NDACC, 2022). MLS v5 data are available from the NASA Goddard Space Flight Center Earth Sciences Data and Information Services Center (GES DISC): https://doi.org/10.5067/Aura/MLS/DATA2516. MERRA-2 data are provided by NASA at the Modeling and Assimilation Data and Information Services Center (MDISC) and are available at the model level (GMAO, 2015a) at https://doi.org/10.5067/WWQSXQ8IVFW8 and in pressure level (GMAO, 2015b) at https://doi.org/10.5067/QBZ6MG944HW0.

*Competing interests.* The contact author has declared that none of the authors has any competing interests.

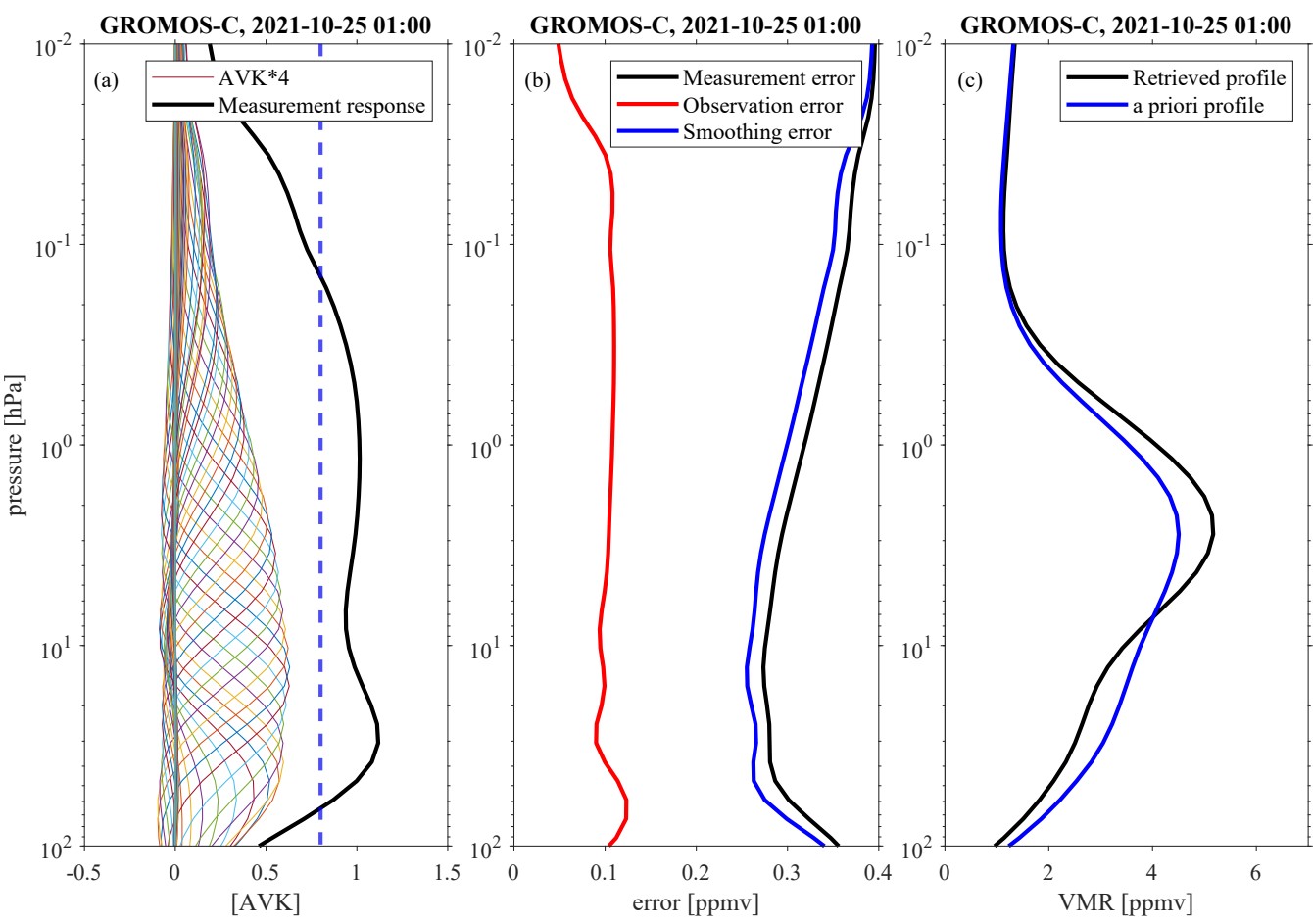

**Figure A1.** Example of GROMOS-C hourly ozone retrievals on 25 October 2021 around 01:00 UT: (a) shows the averaging kernels together with measurement response, (b) shows the measurement, smoothing, and observation errors, (c) shows the retrieved and a priori ozone profile.

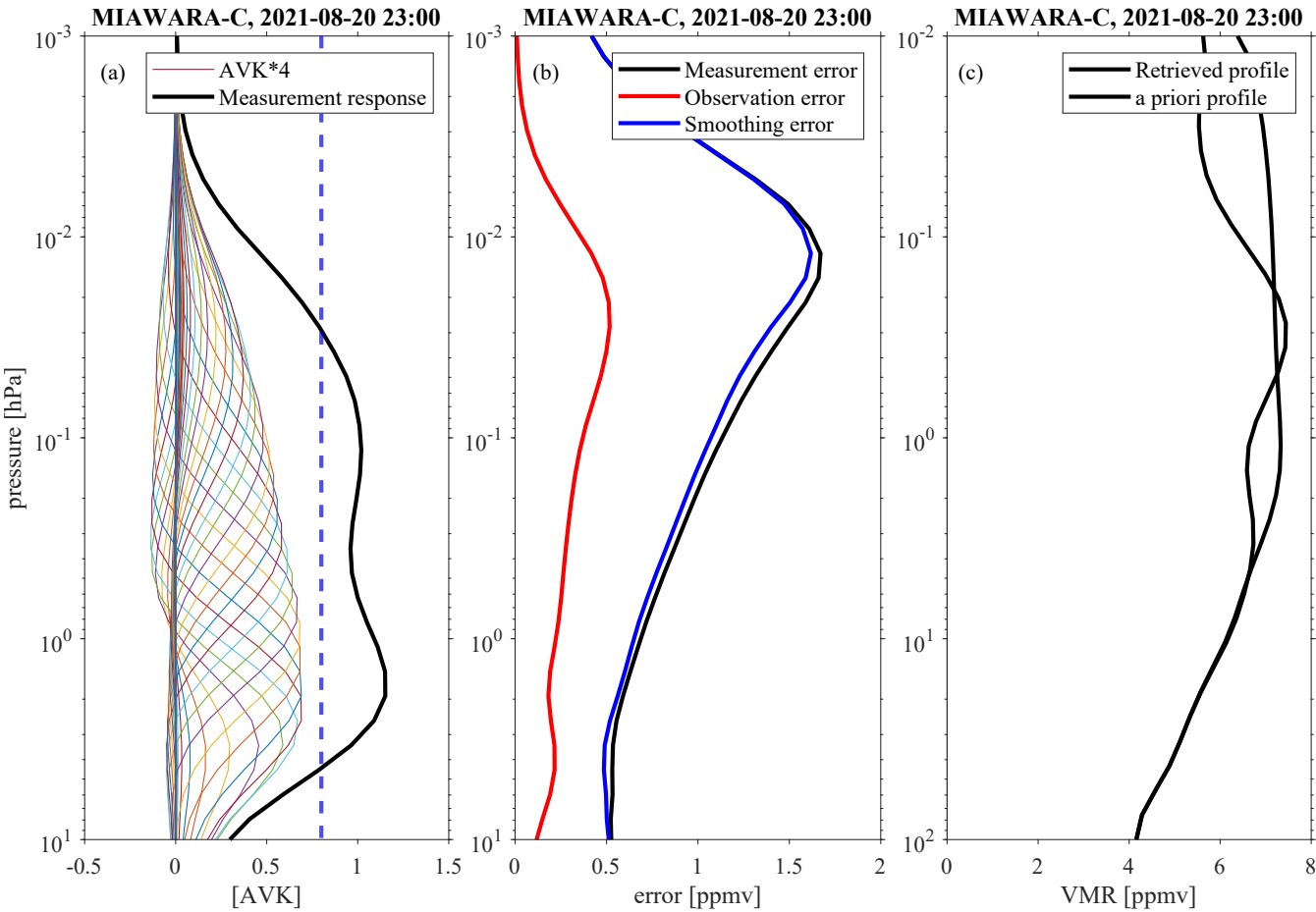

**Figure A2.** Example of MIAWARA-C hourly water vapor retrievals on 20 August 2021 around 23:00 UT: (a) shows the averaging kernels together with measurement response, (b) shows the measurement, smoothing, and observation errors, (c) shows the retrieved and a priori water vapor profile.

## Appendix B: Ozone and water vapor

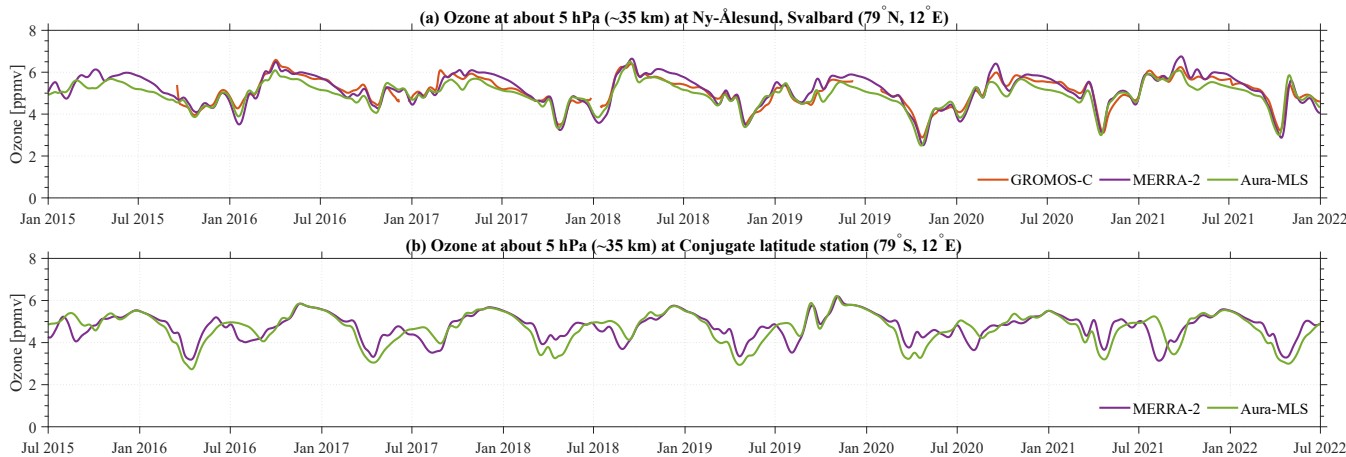

**Figure B1.** GROMOS-C, MERRA-2, and Aura-MLS ozone VMR time series at about 5 hPa smoothed by a 30-day running mean over Ny-Ålesund, Svalbard (79° N, 12° E) and conjugate latitude station (79° S, 12° E).

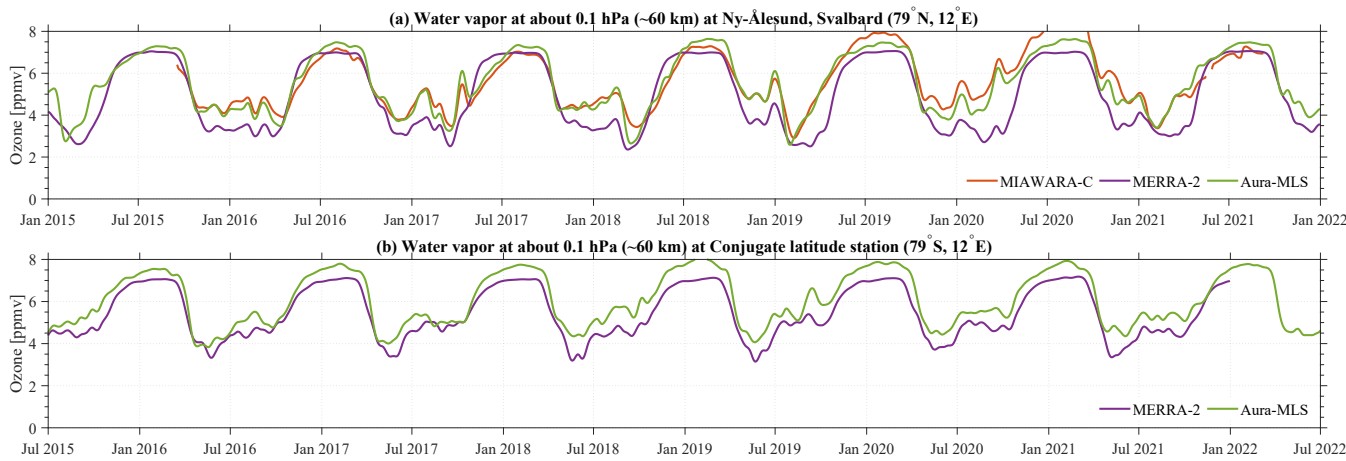

**Figure B2.** MIAWARA-C, MERRA-2, and Aura-MLS water vapor VMR time series at 0.1 hPa smoothed by a 30-day running mean over Ny-Ålesund, Svalbard (79° N, 12° E) and conjugate latitude station (79° S, 12° E).

## Appendix C: Time series of temperature at the mid and polar latitudes

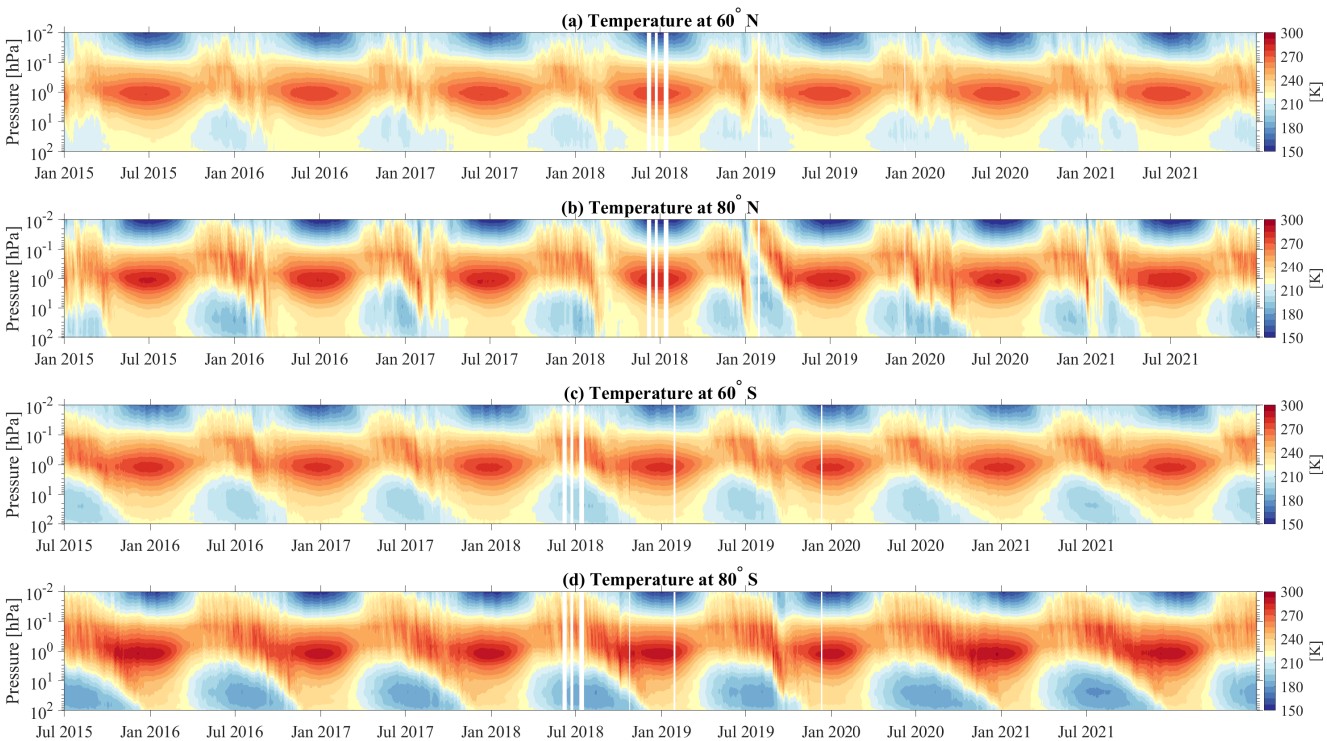

**Figure C1.** Time series of temperature from Aura-MLS observations as a function of pressure over conjugate polar latitudes (80° N & 80° S) and conjugate mid-latitudes (60° N & 60° S) in both hemispheres.

*Author contributions.* GS was responsible for the ground-based ozone measurements with GROMOS-C, performed the data analysis, and prepared the manuscript. ES provided the Aura-MLS data. WK helped with data analysis. GS designed the concept of the study and contributed to the interpretation of the results. All of the authors discussed the scientific findings and provided valuable feedback for manuscript editing.

*Acknowledgements.* The authors acknowledge NASA Global Modeling and assimilation Office (GMAO) for providing the MERRA-2 re-analysis data and the Aura/MLS team for providing the satellite data. We thank the Swiss Polar Institute (SPI) and the Institute of Applied Physics (IAP) supports the development of the GROMOS-C and MIAWARA-C radiometers.

*Financial support.* This research has been supported by the Swiss National Science Foundation (grant no. 200021-200517/1).

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
