# Peer review of "Ozone and water vapor variability in the polar middle atmosphere observed with ground-based microwave radiometers"

_EGUsphere, 2023_

## Referee Comment (RC1)

**Journal:** egusphere

**Title:** Ozone and water vapor variability in the polar middle atmosphere observed with ground-based microwave radiometers

**Author(s):** Guochun Shi et al.

**MS No.:** egusphere-2023-149

**MS type:** Research article

**Special Issue:** Atmospheric ozone and related species in the early 2020s: latest results and trends (ACP/AMT inter-journal SI)

**Review**

The paper reports observations of ozone and water vapour (WV) in the stratosphere and mesosphere above Ny-Ålesund, Svalbard using two ground-based microwave radiometers (GROMOS-C and MIAWARA-C). The ~six-year datasets are compared with overlapping satellite observations from the Aura MLS instrument and reanalysis data (MERRA-2). The datasets are used to investigate interhemispheric differences in ozone and WV, with the Arctic location (79 °N, 12 °E) showing greater interannual and seasonal variability in volume mixing ratio (VMR) for both trace gases than at the corresponding southern hemisphere (SH) location (79 °S, 12 °E) above Antarctica. A case is made for using WV profiles as a tracer of vertical transport associated with meridional (residual) circulation, where upwelling occurs above the summer pole and air typically descends in the winter polar vortex. Estimated vertical velocities from the ground-based radiometer (MIAWARA-C) and MLS WV VMR profiles indicate higher variability in atmospheric dynamics and transport at high northern latitudes compared to the SH.

Ozone and WV are important trace gases in the stratosphere and mesosphere, involved in chemical, radiative, and dynamical transport processes. In the polar regions, at latitudes above 60 °N and 60 °S, these species show seasonal variability due to solar UV, atmospheric circulation, polar stratospheric and mesospheric cloud formation, and impacts from space weather and energetic particle precipitation. Understanding these processes requires observations of ozone and WV, which can be made from the ground, balloon, and satellite remote sensing. This paper makes a useful contribution by reporting a new dataset derived from ground-based observations which is compared with satellite data from Aura MLS and meteorological reanalysis data. As the authors point out, satellites measuring middle atmospheric composition are reaching the end of their lifetimes, and ground-based observations such as these complement and extend the long-term datasets.

The main topic areas of the paper – ozone and WV and atmospheric circulation / transport– appear suitable for the journal and special issue. The introduction section is reasonably well written, outlining the relevance of ozone and WV in the polar middle atmosphere and measurement techniques. The results and discussion contain sufficient detail and follow a logical order with fairly clear conclusions. However, the methodology and analysis lack a discussion of measurement errors and uncertainties which is important when comparing datasets. The plotted figures can also be improved. The main issues are expressed in my three major comments. I have identified numerous other areas in the text where improvements and clarifications are needed. My minor comments and suggested edits are listed below the major comments. In conclusion, I recommend a major revision of the paper, with the authors addressing my main comments and each minor point, before the paper is considered further for publication in *egusphere.*

**Major comments**

1. When comparing datasets from different measurements, it is important to consider measurement and retrieval uncertainties and, for microwave instruments, the choice of a priori data and the effect of limited altitude resolution. Uncertainties in the ozone and WV VMRs from the radiometers and Aura MLS need to be given and considered for example when looking at differences between the observations and reanalysis data. How do the measurement

uncertainties impact on the estimated vertical transport rates? For the microwave instrument retrievals, what effect do the averaging kernels and a priori data have on the retrieved profiles?

2. Although it is suggested (Lines 198–199) that the conjugate latitude station results are lagged by six months relative to those for Ny-Ålesund, that is not the case for the SH plots. It would be very helpful to show the lagged plots. Replotting Figs 4–9 with six-month lags on the SH data will more clearly show the hemispheric differences in seasonal ozone and WV behaviour described at length in the text. The dates or months on the axes need to be adjusted accordingly.

3. Increased ozone abundance at 0.03 hPa / 70 km during winter is incorrectly assigned to the secondary ozone layer (Lines 163–164 and 238–240). The VMR increases are more likely due to the formation of the seasonal tertiary ozone layer which occurs at high latitudes during winter at ~0.03–0.02 hPa / 70–75 km. The secondary ozone layer occurs higher up at ~$10^{-2}$ hPa / 96 km, above the useful range of the datasets presented. As well as making this correction, references to previous tertiary ozone observations need to be added.

Minor comments

**Abstract**

- Lines 2–6. The two sentences starting 'Leveraging GROMOS-C…' could be shortened or combined, removing repetition.

- Line 3. 'long-term behavior'. Suggest removing this as six years' observations is not particularly long-term, and the focus of the study is the interannual variability of ozone and WV rather than long-term trends.

- Lines 6–7. 'Overall differences… ozone climatology are on the order of 10–15%...'. Rather than 'climatology', state the quantity (volume mixing ratio?) being referred to. Also, can a more precise range be given rather than 'on the order of…'?

**1 Introduction**

- Line 26. '…summer mesopause temperature up to 100 K away from the…' would be more clearly written as '…summer mesopause temperatures up to 100 K below the…'

- Line 30. '…circulation branch underneath the residual circulation…'. The Brewer Dobson circulation is not immediately beneath the residual circulation. It might be better to write '…circulation branch at altitudes lower than the residual circulation…'

- Line 31 onwards. Brewer-Dobson circulation (BDC) should be defined and the usual abbreviation 'BDC' then used.

- Lines 31–32. 'The Brewer-Dobson Circulation as the major transport pattern in the stratosphere explains the variability of ozone and water vapor.' BDC is a factor in stratospheric ozone and WV variability but, as mentioned later in the manuscript, polar stratospheric clouds can have significant seasonal impacts on ozone and WV abundances in the lower stratosphere.

- Lines 38–39. '…at stratospheric altitudes in the hemispheric winter…' Either give the altitude range or change to e.g., '…in the hemispheric winter stratosphere…'.

- Line 47. '…with the differences…'. Differences in what quantity? Ozone VMR?

- Line 52. '…the air masses dynamic.' Clarify what is meant here.

- Line 55. 'Its distribution…'. State what 'Its' is.

- Line 61. …'column densities, but lack the vertical information.' Column densities are vertical information. Suggest change to '…column densities, but lack vertically-resolved information.'

- Line 68. 'under all weather conditions with a high time resolution of the order of hours except during rain.' Does precipitating or blowing snow affect the observations? A time resolution of

hours might not be considered particularly high – suggest remove the word 'high'. Do rainy conditions mean that observations cannot be made or that they take longer than hours?

- Line 69. 'It can be specially designed…' In what way is a microwave radiometer 'specially designed' to measure ozone and WV?

- Line 70. '…and provides datasets'. Suggest remove these three words.

- Lines 70–72. The sentence 'Microwave radiometers can continuously measure the middle atmospheric ozone and water vapor which is valuable as it complements satellite measurements are relatively easy to maintain, and has a long lifetime.' should probably be 'Microwave radiometers can continuously measure middle atmospheric ozone and water vapor, which is valuable as it complements satellite measurements, and they are relatively easy to maintain and have long lifetimes.' How long is a radiometer lifetime?

- Lines 72–73. 'Ground-based microwave radiometry is the ideal technique to monitor ozone and water vapor in the Arctic/Antarctic middle atmosphere.' Why is this 'the ideal technique' for these measurements?

**2.1 GROMOS-C**

- Line 91. Suggest change 'GROMOS-C is designed to be very compact' to 'GROMOS-C is very compact'.

- Line 93. Define 'CO'.

- Lines 95–96. Were ozone profiles retrieved for each of the four cardinal directions or a combined retrieval of the ozone profile vertically above the instrument? How does the GROMOS-C observation geometry compare to the Aura-MLS sampling (Lines 122–123)? Presumably radiometric calibration is performed to obtain ozone spectra. Why is it necessary to average 2 hours data?

- Line 97. Define the abbreviation 'ARTS'. 'QPACK' should be 'Qpack' (also on Line108).

**2.2 MIAWARA-C**

- The observation geometry (azimuthal direction and elevation) needs to be stated and, as for GROMOS-C, compared with the Aura-MLS sampling (Lines 122–123).

- Line 105. 'The signal'. What signal is being referred to here?

- Line 110. Why is it necessary to average 2-4 hours' data?

**2.3 Aura-MLS**

Line 113. Define 'EOS'.

Line 115. Suggest change 'MLS scans the limb…' to 'MLS scans the atmospheric limb…'.

Line 116. '…spaced 1.5 degrees…'. 1.5° in latitude or 1.5° in longitude?

Line 122. Change 'two times a day' to 'twice a day'.

Line 122. 'location is within ±400 km latitude and ±800 km longitude'. The units of latitude and longitude should be degrees (°). Either give the ± latitude and longitude ranges in ° or change wording to indicate ± km north–south and ± km east–west of Ny-Ålesund.

**2.4 MERRA-2**

- Line 131. 'We use the ozone and water vapor with 72 hybrid-eta levels…'. What does 'hybrid-eta' mean? State whether MERRA-2 model level or pressure level datasets were used.

**3  Time series of ozone and water vapor**

- Line 135.  'GROMOS-C and MIAWARA-C perform highly accurate continuous ozone and water vapor measurements…'.  Why are the measurements 'highly accurate'?  To back this up, statements on the measurement accuracies of each instrument are needed (see also Major comment 1).  Also, although the instruments have the potential to make continuous measurements, the data gaps in the plots (e.g., Figure 2(a)) suggest the measurements are at best 'near continuous'.  What are the reasons for the measurement data gaps?

**3.1  Ny-Ålesund, Svalbard (79 °N, 12 °E) in the NH**

- Line 151.  '…photo-chemically production…' should be '…photochemical production…'.

- Lines 161–162.  'During late summer and early autumn, stratospheric ozone decreases rapidly when the vortex passes over Ny-Ålesund.'  This is incorrect: - the polar vortex occurs in wintertime, not late summer and early autumn.

- Line 163.  '…the very big values…'.  State the values (see also Major comment 3).

- Line 185.  '…water vapor VMR reaches the maximum at 10 hPa.'  The 10 hPa pressure level is below the recommended retrieval measurement response (MR) > 0.8 so how well can these values be trusted?  Presumably where MR is below 0.8 the radiometer retrievals are driven primarily by the a priori profile.  The a priori data used in retrievals from each instrument needs to be stated.

- Line 186.  'In some years,…'.  State the years.

- Line 192.  'MERRA2' should be 'MERRA-2' for consistency.

**3.2  Conjugate latitude  station (79 °S, 12 °E) in the SH**

- Lines 198–199.  'For comparison purposes, the conjugate latitude station results are lagged by 6 months relative to those for Ny-Ålesund.'  The SH plots in Figs 4-9 are not seasonally lagged but it is recommended that is done (see Major comment 2).

- Line 200.  '…ozone VMR maximum of about 6 ppmv in winter and a minimum of about 4 ppmv in summer at about 5 hPa…'.  The words 'summer' and 'winter' need to be swapped round.

- Line 207.  '…models…' should be '…MERRA-2 data…'.

- Line 211.  '…and irreversibly remove water and nitric acid from the upper atmosphere…'.  Unclear what the upper atmosphere (i.e., thermosphere and above) has to do with this: - suggest remove the words 'from the upper atmosphere'.

**4  Climatologies of ozone and water vapor**

- Line 216.  'It is important…'  Why is it important?

**4.1  Ozone**

- Line 227.  '…maximum observed and modeled in the SH (approximately 5.5 ppmv) is somewhat smaller and earlier in the season…'  Suggest change 'maximum observed and modeled' to 'maximum observed and reanalysis ozone VMR'.  Also, smaller by how much, earlier by how much, which season?

- Line 232.  'Polar cap' usually means above the Antarctic land mass or at very high latitude. Suggest change to 'polar region'.

- Line 236.  'The SH…' should be 'The SH ozone VMR…'

**4.2 Water vapor**

- Line 249. '…relatively long photo-chemical lifetime of water vapor, …'.  What is the lifetime?

- Line 251. '…monthly water vapor VMR…' should probably be '…monthly mean water vapor VMR…'.

**4.3 Relative differences**

- Line 271–279.  It would be helpful in the text discussion for this section to refer to the specific panels of Fig. 10 e.g., Fig. 10(a) on Line 271 and Figs 10(b-d) at the relevant places for these panels.

**5  Dynamics and transport of water vapor**

- Line 300. '…regression fit to different water vapor mixing ratio isopleths…'.  State all the isopleth values used here.

- Line 301. 'The time period of 7 years…'  For MIAWARA-C the time period appears to be ~six years (2015–2020).

- Line 303. '…dynamic and chemical…'.  Dynamic and chemical what?

- Line 309. '…the ascent rate in 2019 and 2020 is slightly earlier…'  Rather than 'slightly earlier', quantify how much earlier.

- Line 317.  Suggest change '…similar for the 7 years, likely due to the higher stability and stronger of the southern polar vortex.' to '…similar for each of the seven years, likely due to the higher stability and strength of the southern polar vortex.'

- Lines 322–325 and Table 1 (also Discussion and Conclusions sections, also see Major comment 1).  The vertical velocities are given to three significant figures, but I wonder whether the linear regression fits are that significant.  Would two significant figures suffice?  Also, the values only need to be presented once, preferably in Table 1, and discussed in the text.  Do the vertical velocities from fits to the different isopleths differ?

**6  Discussion**

- Lines 350. '…Aura-MLS observations…'.  Clarify whether the Aura-MLS temperatures are zonally averaged.  How much colder are the southern polar latitudes?

- Line 354. 'There are only a few occasions…'  When were these occasions?

- Lines 363–364. 'During this time of the year, horizontal gradients…'  What time of year is being referred to here?  Horizontal gradient in what?

- Line 380. '…hydroxyl OH…' should be '…hydroxyl (OH)…'.

**7  Conclusions**

- Lines 388 and 401.  'Ny-Åesund' should be 'Ny-Ålesund'.

- Line 389.  'AURA' should be 'Aura'.

- Line 393.  'satellite observations such as MLS…' should be 'satellite instruments such as MLS…'

- Lines 407 and 409.  Mention of the quasi-biennial oscillation (QBO) and climate change, and their relevance to this study, should probably be moved to the Introduction.

---

## Author Comment (AC2)

**Response to Reviewer – 'Ozone and water vapor variability in the polar middle atmosphere observed with ground-based microwave radiometers' by Guochun Shi et al.**

We'd like to thank the Reviewer for his/her positive feedback and valuable comments. Here we address the comments of the Reviewer, with his/her comments in black and our responses indented in blue.

In this paper, time-series of stratospheric and mesospheric ozone and water vapor profiles observed from the high-latitude station Ny Alesund with the two ground-based Microwave (MW) radiometers MIAWARA-C and GROMOS-C are discussed. Time-series and climatologies are compared against MLS/AURA satellite observations and MERRA-2 reanalysis data, and the annual and interannual variability is discussed for Ny Alesund based on all three data-sets, and for a "conjugate station" at high SH latitudes based on MLS and MERRA-2 data. Ascending and descending rates in summer respectively winter are derived for the NH from the ground-based MW observations and MLS, for the SH for MLS only. MLS and the ground-based instruments agree very well, both regarding the absolute values up to the mid-mesosphere and the day-to-day variability, while MERRA-2 shows systematic differences to both the ground-based MW observations and MLS in the mesosphere, and to MLS in the SH lower stratospheric ozone hole area. However, considering the low vertical resolution of the ground-based MW observations, it might be more appropriate to degrade MLS and MERRA-2 data to the vertical resolution of MIAWARA-C respectively GROMOS-C before computing the relative differences. However, the good agreement with MLS data demonstrates a high quality of the MIAWARA-C and GROMOS-C instruments, and ground-based observations like these will be of increasing importance in the near future when the last remaining satellite limb observing instruments have reached the end of their lifetime, to continue long-term time-series at least locally, to constraint models like MERRA-2, and hopefully to provide a connection to future limb satellite missions, though the only limb sounder mission that I know of with a coverage comparable to MLS is the Earth Explorer 11 candidate mission CAIRT, with EE11 expected to launch in the early 2030s. In this sense, the presentation of the MIAWARA-C and GROMOS-C data, their evaluation against MLS, and discussion of properties derivable from these data, are very valid and important results adequate for publication in ACP (or AMT).

However, in its current stage, it was not clear to me what the main focus of the paper is – evaluation of the data, derivation of stratospheric annual and interannual variability, interhemispheric differences? – as all these aspects are somewhat mixed together. In my opinion, the paper could become much more clear and concise by restructuring it in such a way that the different aspects are clearly separated. As the evaluation of the MIAWARA-C and GROMOS-C data is an important prerequisite for using them to derive annual/interannual and interhemispheric variabilities, this should be discussed first. Results concerning the interhemispheric differences in the dynamical variability of the stratosphere mainly confirm previous results; what is new here is the instruments used, as well as the concise derivation of descent and ascent rates in both hemispheres from the climatological annual variation of H2O. This is an interesting comparison, though I don't really follow the argument that this is evidence for interhemispheric coupling; this point needs a more detailed explanation.

We appreciate your insightful comments regarding the clarity and organization of the manuscript. We agree that the current manuscript can be improved to highlight the main focus better and separate the different aspects of our study. We understand that the current combination of data evaluation, derivation of stratospheric annual and interannual variability, and interhemispheric differences may have resulted in a lack of clarity. In the revised manuscript, we will discuss the data evaluation process prominently in the initial sections of the paper, then, clearly explain annual and interannual variability and a more detailed explanation of the comparison of descent and ascent rates derived from the climatological annual variation of water vapor. We will give a more detailed explanation of the evidence for interhemispheric coupling and expand upon this aspect in the revised manuscript, providing a more comprehensive explanation of the relationship and its implications.

**A few minor issues are listed below.**

Abstract: state altitude / pressure range of investigation, as well as time-period (2015-2021?) and the vertical range and times of year over which the descent/ascent ranges have been determined.

Changed the sentence as follows:

In the northern hemisphere, the water vapor ascent rate (05 May to 20 Jun from 2015 to 2018, and 15 Apr to 31 May from 2019 to 2021) is 3.4±1.9 mm s−1 from MIAWARA-C and 4.6±1.8 mm s−1 from Aura-MLS, and descent rate(15 Sep-31 Oct from 2015-2021) is 5.0±1.1 mm s−1 from MIAWARA-C and 5.4±1.5 mm s−1 from Aura-MLS at the altitude range of about 50-70 km. The water vapor ascent (15 Oct to 30 Nov from 2015-2021) and descent rates (15 Mar to 30 Apr from 2015-2021) in the southern hemisphere are 5.2±0.8 mm s−1 and 2.6±1.4 mm s−1 from Aura-MLS, respectively.

Line 37: The meridional transport of trace gases into the polar cap is controlled by the strength of the polar vortex …. Add: during polar winter.

Changed the sentence as follows:

The meridional transport of trace gases into the polar cap is controlled by the strength of the polar vortex during polar winter, which is driven by the temperature gradient between the polar cap and the mid-latitudes through the thermal wind balance in the hemispheric winter stratosphere, and forms an essential barrier separating ozone rich air at the mid-latitudes from ozone depleted air within the polar cap.

Sentences in lines 69-70 and 70-71 have partly redundant information, and could be shortened accordingly.

Changed the sentence as follows:

It can be specially designed for measuring ozone and water vapor which is valuable as it complements satellite measurements, is relatively easy to maintain and has a long lifetime, and is operated from different locations on a campaign basis (Scheiben et al., 2013, 2014).

Line 74-76: please clarify in this sentence that MLS data are used to provide the observations for the conjugate station.

Changed the sentence as follows:

Here, we present a detailed comparison of ozone and water vapor observed by Aura-MLS at conjugate latitude station leveraging multiyear ground-based observations from GROMOS-C and MIAWARA-C performed at Ny-Ålesund and Aura-MLS data as well as reanalysis data.

Sections 2.1, 2.2, 2.3: Instrument descriptions: please add from when to when data are available, as well as the instruments precision/noise error, to the descriptions of GROMOS-C, MIAWARA-C, and MLS

Changed the sentence as follows:

In this study, we use ozone and water vapor measurements from our two ground-based microwave radiometers GROMOS-C and MIAWARA-C which are only available to measure at single locations and, thus, are representative of a specific geographic location. Both instruments are located at Ny-Ålesund, Svalbard (79◦ N, 12◦ E) and collected continuous data since September 2015. We extract the interannual ozone and water vapor variability between Jan 2015 and Dec 2021 from MERRA-2 and Aura-MLS over the northern polar station, between Jul 2015 and Jul 2022 from MERRA-2 and Aura-MLS over the southern polar station. The corresponding virtual conjugate latitude station (79◦ S, 12◦ E) is shown in Fig. 1. Additionally, we use temperature observations from Aura-MLS.

GROMOS-C and MIAWARA-C precision/noise error: please see the reply to the first major comment of reviewer1.

MLS precision/noise error:

The estimated temperature single-profile precisions are 1 K at 100 hPa, 0.5 K at 10 hPa, 0.8 K at 1 hPa, 1 K at 0.1 hPa, 1.2 K at 0.01 hPa, and 2 K at 0.001 hPa.

The estimated ozone single-profile precision reported by MLS Level 2 v1.5 software varies from 0.2 to 0.4 ppmv from the mid-stratosphere to the lower mesosphere.

The estimated water vapor single-profile precision reported by MLS Level 2 v1.5 software is 0.2–0.3 ppmv in most of the stratosphere and increases to 0.7-0.8 ppmv in the middle mesosphere.

Section 2.4, MERRA-2 description: please add description how ozone and water vapor information is derived within MERRA-2. Data assimilation (which altitudes), photochemical model (how detailed)? This information is important to understand the deficiencies of MERRA-2 compared to the observations shown in later sections, and should already be provided here.

We agree and as suggested by the reviewer, therefore, we added the description how ozone and water vapor information within MERRA-2 in Section 2.4.

Line 154: Fig. 2 shows that the ozone VMR starts to increase …

Changed the sentence as follows:

Fig.2 shows that the ozone VMR starts to increase up to the maximum value of about 8 ppmv for some of the years in the stratosphere when the polar vortex is disturbed or weakened by the planetary waves leading to the formation of sudden stratospheric warming events.

Lines 163-164: this sentence appears to end in the middle

Changed this sentence appears to end in the middle.

Lines 164-165, 169-170: I think these details about ozone in the MERRA-2 dataset should be presented in the description of MERRA-2 in Section 2.3 already. That would make the argument here more simple – it seems to be a known and well-understood bias in MERRA (at least it is consistent with what I would expect from a model that uses a simplified stratospheric chemistry scheme in the mesosphere).

These sentences have been modified in Section 2.3.

172-173: below 1 hPa. If you look closely, you will note that the annual variation in MERRA-2 ozone is significantly different from the observed variation throughout the mesosphere above 1 hPa.

Changed this sentence as follows:

Remarkably, Fig.2 shows the consistencies of GROMOS-C with both MERRA-2 and Aura-MLS datasets in time series of ozone below 1 hPa. A clear annual cycle in the stratosphere is well captured by all datasets, and the higher variability of ozone in winter and spring seasons is clearly visible.

Line 189-190: erase either the captured, or the agrees

Changed the sentence as follows:

In general, MIAWARA-C the annually varying mesospheric distribution of water vapor agrees well with reanalysis data and satellite observations.

Line 202-213, discussion of ozone and water vapor at conjugate latitudes: the differences in the stratospheric dynamics between NH and SH and its impact on stratospheric ozone is fairly well known,

and well understood. I see two take-away messages from figures 4 and 5: (1) while MERRA-2 generally does a good job in reproducing the variability of stratospheric ozone and water vapor, it fails in the mesosphere as discussed in previous sections, and also underestimates the vertical extent of the ozone hole, which appears to end at lower altitudes (larger pressures) in MERRA-2 than in MLS. This is reflected also in water vapor, where MERRA-2 fails to reproduce the vertical layering (e.g. July 2015 or July 2017), and appears to underestimate the area of dehydration. (2) The detailed information about the SH winter as shown here for the "conjugate station" will be lost after the end of the last three limb sounders still observing (MLS/AURA, SABER, SMR/ODIN), while for the NH, information will hopefully continue to be available from ground-based MW observations.

These messages have been shown in the revised manuscript to provide a more comprehensive analysis. We state the underestimation of the vertical extent of the ozone hole in MERRA-2 compared to MLS. We explore potential reasons for these discrepancies and discuss the implications for understanding stratospheric dynamics and ozone depletion processes. Additionally, we highlight the potential for ground-based microwave observations to continue providing valuable information for the Northern Hemisphere (NH). We discuss the significance of these observations in providing continuous monitoring capabilities and filling the observational gap left by limb sounders.

Lines 217-218: It is important … I do agree with this sentence. However, this evaluation of GROMOS-C and MIAWARA-C should be discussed much earlier in the paper, before the annual/interannual variability and descent rates are discussed.

The evaluation of GROMOS-C and MIAWARA-C have been discussed in section 3 (Climatology of ozone and water vapor). Please see the reply to the major comment.

Sections 4.1, 4.2, and 4.3: In general, it would be beneficial to clearly separate the evaluation of the MW data from the data analysis, and to discuss the evaluation first. That is, first discuss the climatologies and their relative differences in the NH, where MW observations are available. Discuss the difference between NH and SH in the next section, then the time-series, and finally the descent rates.

We agree and as suggested by the reviewer, firstly we discuss the climatologies and their relative differences in the NH in section 3 and the time series of ozone and water vapor in section 4 and ascent and descent rates in section 5.

Section 4.3, discussion of Figure 10: is it possible that some of the differences are due to the low vertical resolution of the ground-based MW instruments? That is, could the agreement between MLS and GROMOS/MIAWARA be even better if the MLS data were convolved to the vertical resolution of the GROMOS/MIAWARA observations, e.g., by applying a convolution with the averaging kernels?

Our revision shows MERRA-2 and MLS are convolved with the averaging kernels of GROMOS-C and MIAWARA-C. The vertical resolution of MW instruments is generally coarser compared to satellite limb sounders like MLS, which can result in some differences when comparing the vertical profiles directly. By convolving the MLS data to the vertical resolution of GROMOS-C/MIAWARA-C, we can obtain a better alignment of the observations and assess the level of agreement in a more comparable manner. In the revised manuscript, we will discuss the potential impact of the vertical resolution differences on the observed discrepancies and address the possibility of convolving MLS and MERRA-2 data with the averaging kernels of GROMOS-C/MIAWARA-C.

Here, we show two figures about the relative difference, without the convolution of the averaging kernels and with the convolution of the averaging kernels.

[Figure]

Lines 372-375: I don't really understand the reasoning here. My understanding of the interhemispheric coupling (based mainly on Körnich and Becker, 2010) is that disturbances are transmitted from the winter stratosphere to the summer mesosphere in the other hemisphere; so a high variability in NH winter downwelling would imply a high variability of SH summer upwelling, which is indeed observed; but also a low variability in the SH winter downwelling would imply a low variability of the NH summer upwelling, however, the converse is observed. I'd say the issue of interhemispheric coupling and how your results relate to that, needs a bit more explanation / discussion.

A high variability in NH winter downwelling would imply a high variability of SH summer upwelling, which is indeed observed in Fig 13. This suggests that the interhemispheric coupling (Körnich and Becker, 2010; Orsolini et al., 2010; Smith et al., 2020a) is that disturbances are transmitted from the winter stratosphere to the summer mesosphere in the other hemisphere. However, it is worth noting that there is a lower variability in SH winter downwelling, which corresponds to a higher variability of NH summer upwelling. This observation suggests that the coupling between the winter and summer

circulations within each hemisphere is influenced by various factors beyond the interhemispheric exchange. As shown in Fig 14, with a stronger temperature gradient during SH winter, the polar vortex is relatively stable and well-defined, leading to reduced variability in the downward motion of air (downwelling). For instance, the SSW events during the NH winter (Limpasuvan et al., 2016; Schranz et al., 2019, 2020) or increased stratospheric planetary wave activity (De Wit et al., 2015) lead to increased variability in the NH winter stratosphere.

Line 378: that results in both what?

Changed the sentence as follows: Simulations with a gravity wave resolving model response to the enhanced winter hemisphere Rossby-wave activity could be the mechanism that results in both an interhemispheric coupling through a downward shift of the GW-driven branch of the residual circulation and an increased GW activity at high summer latitudes (Becker and Fritts, 2006).

---

## Author Comment (AC3)

**Response to Reviewer – 'Ozone and water vapor variability in the polar middle atmosphere observed with ground-based microwave radiometers' by Guochun Shi et al.**

We'd like to thank the Reviewer for his/her positive feedback and valuable comments. Here we address the comments of the Reviewer, with his/her comments in black and our responses indented in blue.

**Major comments**

1. When comparing datasets from different measurements, it is important to consider measurement and retrieval uncertainties and, for microwave instruments, the choice of a priori data and the effect of limited altitude resolution. Uncertainties in the ozone and WV VMRs from the radiometers and Aura MLS need to be given and considered for example when looking at differences between the observations and reanalysis data. How do the measurement uncertainties impact on the estimated vertical transport rates? For the microwave instrument retrievals, what effect do the averaging kernels and a priori data have on the retrieved profiles?

We appreciate your valuable comment regarding the consideration of measurement and retrieval uncertainties when comparing datasets from different measurements, particularly for microwave instruments. We will address these aspects in the revised manuscript to provide a more comprehensive analysis.

The two figures show the averaging kernels and measurement errors and single ozone profile from GROMOS-C and MIAWARA-C. Averaging kernels are a measure of the sensitivity of the retrieval to the true ozone profile at each pressure level. The sum of the AVKs at each level defines the measurement response. It is an indication of the measurement contribution to the retrieved profile, whereas the remaining information comes from the a priori. Also included as diagnostic quantities are the smoothing and measurement errors computed by the optimal estimation method as defined by Rodgers (2000). In microwave remote sensing, the measurement response of 0.8 is often used to define the lower and upper boundaries of the retrievals in order to limit the influence of the a priori on the results. The expected error from GROMOS-C has been estimated at about 0.25-0.4 ppmv. The estimated ozone single-profile precision reported by the Aura-MLS Level 2 software typically varies from 0.2 to 0.4 ppmv (or 2% to 15%) from the mid-stratosphere to the lower mesosphere. The expected error from MIAWARA-C has been estimated at about 0.4-1 ppmv in the upper stratosphere and lower mesosphere and increases to 1.5 ppmv at the mesopause. Typical ozone single-profile precisions reported by Aura-MLS Level 2 v1.5 software are 0.2–0.3 ppmv in most of the stratosphere and increase to 0.7-0.8 ppmv in the middle mesosphere. The vertical velocities should be given to two significant digits (For example, velocity is ~4.6±0.2 mm/s) and the linear regression fits are very significant. Please see Figure 13, the uncertainties for the ascent and descent rates of airflow over northern and southern polar latitude stations are depicted by error bars in Figure 13.

[Figure]

2. Although it is suggested (Lines 198–199) that the conjugate latitude station results are lagged by six months relative to those for Ny-Ålesund, that is not the case for the SH plots. It would be very helpful to show the lagged plots. Replotting Figs 4–9 with six-month lags on the SH data will more clearly show the hemispheric differences in seasonal ozone and WV behaviour described at length in the text. The dates or months on the axes need to be adjusted accordingly.

We appreciate your feedback and suggestion regarding the lagged plots for the conjugate latitude station results in relation to Ny-Ålesund. We will address your comment by replotting Figures 4-9 with the appropriate six-month lags for the SH data. To ensure consistency and improve clarity, we will adjust the dates or months on the axes accordingly to reflect the lag. This adjustment will enable readers to better understand the temporal relationship between the SH data and the corresponding conjugate latitude station results.

3. Increased ozone abundance at 0.03 hPa / 70 km during winter is incorrectly assigned to the secondary ozone layer (Lines 163–164 and 238–240). The VMR increases are more likely due to the formation of the seasonal

tertiary ozone layer which occurs at high latitudes during winter at ~0.03–0.02 hPa / 70–75 km. The secondary ozone layer occurs higher up at ~$10^{-2}$ hPa /96 km, above the useful range of the datasets presented. As well as making this correction, references to previous tertiary ozone observations need to be added.

We will correct three ozone layers.

Lines 163–164: Marsh et al. (2001) and Smith et al. (2009, 2018) combined models and observations to investigate the location of tertiary ozone maximum in the winter mesosphere at high latitudes. In this study, GROMOS-C and Aura-MLS present the seasonal tertiary ozone layer at 0.03–0.02 hPa (about 70–75 km) in winter months, but the tertiary ozone VMRs are very big values above 0.1 hPa in MERRA-2 (as shown in the red shading in Fig. 2b).

References:

Marsh, D., Smith, A., Brasseur, G., Kaufmann, M., and Grossmann, K.: The existence of a tertiary ozone maximum in the high-latitude middle mesosphere, Geophysical Research Letters, 28, 4531–4534, 2001.

Smith, A., López-Puertas, M., García-Comas, M., and Tukiainen, S.: SABER observations of mesospheric ozone during NH late winter 2002–2009, Geophysical Research Letters, 36, 2009.

Smith, A. K., Espy, P. J., López-Puertas, M., and Tweedy, O. V.: Spatial and temporal structure of the tertiary ozone maximum in the polar winter mesosphere, Journal of Geophysical Research: Atmospheres, 123, 4373–4389, 2018.

Lines 238–240: Another essential feature is that GROMOS-C and Aura-MLS capture the tertiary ozone VMR maximum at the northern polar latitude in the early winter and in late spring at the southern polar latitude. Although, the tertiary ozone maximum in GROMOS-C occurs at altitudes close to or even above the limit of 0.8 for the measurement response.

**Minor comments**

**Abstract**

• Lines 2–6. The two sentences starting 'Leveraging GROMOS-C…' could be shortened or combined, removing repetition.

Changed the sentence as follows:

Leveraging continuous ozone and water vapor measurements with the two ground-based radiometers GROMOS-C and MIAWARA-C at Ny-Ålesund, Svalbard (79°N, 12°E) that started in September 2015 and combining MERRA-2, and Aura-MLS datasets, we analyze the interannual behavior and differences of ozone and water vapor and compile climatologies of both trace gases that describe the annual variation of ozone and water vapor at polar latitudes.

• Line 3. 'long-term behavior'. Suggest removing this as six years' observations is not particularly long-term, and the focus of the study is the interannual variability of ozone and WV rather than long-term trends.

Changed the sentence as follows:

Leveraging continuous ozone and water vapor measurements with the two ground-based radiometers GROMOS-C and MIAWARA-C at Ny-Ålesund, Svalbard (79°N, 12°E) that started in September 2015 and combining MERRA-2, and Aura-MLS datasets, we analyze the interannual behavior and differences of ozone and water vapor and compile climatologies of both trace gases that describe the annual variation of ozone and water vapor at polar latitudes.

• Lines 6–7. 'Overall differences… ozone climatology are on the order of 10–15%...'. Rather than 'climatology', state the quantity (volume mixing ratio?) being referred to. Also, can a more precise range be given rather than 'on the order of…'?

Changed the sentence as follows:

Overall differences between GROMOS-C and Aura-MLS ozone (volume mixing ratio) climatology are 10-15 % depending on the altitudes.

**1 Introduction**

• Line 26. '…summer mesopause temperature up to 100 K away from the…' would be more clearly written as '…summer mesopause temperatures up to 100 K below the…'

Changed the sentence as follows:

Model results suggest that gravity waves drive the summer mesopause temperature up to 100 K below the radiative equilibrium (Lindzen, 1981; Smith, 2012; Becker, 2012).

• Line 30. '…circulation branch underneath the residual circulation…'. The Brewer Dobson circulation is not immediately beneath the residual circulation. It might be better to write '…circulation branch at altitudes lower than the residual circulation…'

Changed the sentence as follows:

Another important circulation branch at altitudes lower than the residual circulation is the Brewer-Dobson Circulation (BDC) (Brewer, 1949; Dobson, 1956).

• Line 31 onwards. Brewer-Dobson circulation (BDC) should be defined and the usual abbreviation 'BDC' then used.

Changed the sentence as follows:

BDC is much weaker during boreal summer due to the different distribution of land masses and the associated differences in the generation of planetary and gravity waves between both hemispheres. BDC can govern the entry and distribution of air masses and constituents from the troposphere into and within the stratosphere.

• Lines 31–32. 'The Brewer-Dobson Circulation as the major transport pattern in the stratosphere explains the variability of ozone and water vapor.' BDC is a factor in stratospheric ozone and WV variability but, as mentioned later in the manuscript, polar stratospheric clouds can have significant seasonal impacts on ozone and WV abundances in the lower stratosphere.

Changed the sentence as follows:

BDC as the major transport pattern in the stratosphere has a significant impact on ozone and water vapor distribution.

• Lines 38–39. '…at stratospheric altitudes in the hemispheric winter…' Either give the altitude range or change to e.g., '…in the hemispheric winter stratosphere…'.

Changed the sentence as follows:

The meridional transport of trace gases into the polar cap is controlled by the strength of the polar vortex, which is driven by the temperature gradient between the polar cap and the mid-latitudes through the thermal wind balance in the hemispheric winter stratosphere, forms an essential barrier separating ozone rich air at the mid-latitudes from ozone depleted air within the polar cap.

• Line 47. '…with the differences…'. Differences in what quantity? Ozone VMR?

Changed the sentence as follows:

Because most ozone is found in the lower stratosphere, the differences in the column ozone distribution explain the asymmetry because of dynamic transport, as well as the interannual variability of ozone in both hemispheres (McConnell and Jin, 2008; Langematz, 2019).

• Line 52. '…the air masses dynamic.' Clarify what is meant here.

Changed the sentence as follows:

Water vapor has a chemical lifetime of the order of months in the upper stratosphere and lower mesosphere (Brasseur and Solomon, 2005), therefore, it can be used as a tracer to study a large-scale upwelling and downwelling of the air masses in the polar mesosphere.

• Line 55. 'Its distribution…'. State what 'Its' is.

Changed the sentence as follows:

The distribution and variability of ozone and water vapor exhibit a wealth of information on atmospheric circulation.

• Line 61. …'column densities, but lack the vertical information.' Column densities are vertical information. Suggest change to '…column densities, but lack vertically-resolved information.'

Changed the sentence as follows:

Ground-based observations are often performed using Brewer and Dobson instruments (Zuber et al., 2021), which provide very high quality and precision ozone column densities, but lack the vertically-resolved information. Lidars are providing good vertical resolution to measure ozone (Brinksma et al., 1997; Bernet et al., 2021).

• Line 68. 'under all weather conditions with a high time resolution of the order of hours except during rain.' Does precipitating or blowing snow affect the observations? A time resolution of hours might not be considered particularly high – suggest remove the word 'high'. Do rainy conditions mean that observations cannot be made or that they take longer than hours?

Precipitating or blowing snow can affect the observations. Precipitation or snow can increase the noise figure by so much that GROMOS-C is no longer sensitive enough to observe the water emissions from the stratosphere and mesosphere.

The word 'high' has been removed and the sentence has been changed as follows:

The ground-based microwave radiometer allows a continuous observation under all weather conditions with a time resolution of the order of hours except during rain.

• Line 69. 'It can be specially designed…' In what way is a microwave radiometer 'specially designed' to measure ozone and WV?

Both instruments are specially designed for campaigns and, thus, they measure autonomously and only need power and an internet connection.

• Line 70. '…and provides datasets'. Suggest remove these three words.

These three words have been removed and the sentence has been changed as follows:

It can be specially designed for measuring ozone and water vapor and operated from different locations on a campaign basis (Scheiben et al., 2013, 2014).

• Lines 70–72. The sentence 'Microwave radiometers can continuously measure the middle atmospheric ozone and water vapor which is valuable as it complements satellite measurements are relatively easy to maintain, and has a long lifetime.' should probably be 'Microwave radiometers can continuously measure middle atmospheric ozone and water vapor, which is valuable as it complements satellite measurements, and they are relatively easy to maintain and have long lifetimes.' How long is a radiometer lifetime?

Changed the sentence as follows:

It is specially designed for measuring ozone and water vapor which is valuable as it complements satellite measurements, is relatively easy to maintain, and has a long lifetime which ensures a long and continous time series that can cover several decades

The lifetime of a radiometer can vary depending on several factors, including the specific instrument, its design, the quality of components, and maintenance practices. Generally, a well-maintained and regularly calibrated radiometer can have a useful lifetime of several years to a decade or more.

• Lines 72–73. 'Ground-based microwave radiometry is the ideal technique to monitor ozone and water vapor in the Arctic/Antarctic middle atmosphere.' Why is this 'the ideal technique' for these measurements?

Ground-based microwave radiometer is that they measure continuously day and night in poles and even under cloudy conditions with a time resolution of up to 30 minutes. The instruments operate autonomously which allows measurements at remote places and are relatively easy maintenance and have long lifetimes. Ground-based radiometry complements satellite measurements by providing independent and concurrent observations of ozone and water vapor in the middle atmosphere.

**2.1 GROMOS-C**

• Line 91. Suggest change 'GROMOS-C is designed to be very compact' to 'GROMOS-C is very compact'.

Changed the sentence as follows:

GROMOS-C is very compact so it can be transported and operated at remote field sites under extreme climate conditions.

• Line 93. Define 'CO'.

Defined 'CO' as carbon monoxide.

• Lines 95–96. Were ozone profiles retrieved for each of the four cardinal directions or a combined retrieval of the ozone profile vertically above the instrument? How does the GROMOS-C observation geometry compare to the Aura-MLS sampling (Lines 122–123)? Presumably radiometric calibration is performed to obtain ozone spectra. Why is it necessary to average 2 hours data?

Ozone profiles were retrieved for each of the four cardinal directions. At an altitude of 3 hPa (37 km) the distance between the E/W and N/S measurement locations is 184 km. Profiles for comparison are extracted if the location is within ±1.2∘ latitude and ±6∘ longitude of Ny-Ålesund. Averaging 2 hours of data are required to reduce the noise in the spectra to ensure a robust retrieval of the ozone.

• Line 97. Define the abbreviation 'ARTS'. 'QPACK' should be 'Qpack' (also on Line108).

Changed the sentence as follows:

Ozone volume mixing ratio (VMR) profiles are retrieved from the ozone spectra with a temporal averaging of 2 hours leveraging Atmospheric Radiative Transfer Simulator version2 (ARTS2; Eriksson et al., 2011) and Qpack2 software (Eriksson et al., 2005) according to the optimal estimation algorithm (Rodgers, 2000).

2.2 MIAWARA-C

• The observation geometry (azimuthal direction and elevation) needs to be stated and, as for GROMOS-C, compared with the Aura-MLS sampling (Lines 122–123).

Added the sentences as follows:

MIAWARA-C is pointing to an azimuth of 18∘ east of north. Every 15 minutes the ambient load is measured for about 2 s and the sky at 60∘ elevation is measured for about 15 s.

• Line 105. 'The signal'. What signal is being referred to here?

'The signal' is the polarised signal.

Changed the sentences as follows:

The antenna is followed by a dual polarization receiver. The incident radiation is split into vertical and horizontal polarisation by an orthomode transducer (OMT) placed immediately after the feedhorn. The signal is split into two polarizations by an orthomode transducer directly. The two polarised signals are processed in the two identical receiver chains and separately analyzed in a fast Fourier transform (FFT) spectrometer model Acqiris AC240.

• Line 110. Why is it necessary to average 2-4 hours' data?

Averaging 2 hours of data are required to reduce the noise in the spectra to ensure a robust retrieval of the water vapor.

2.3 Aura-MLS

Line 113. Define 'EOS'.

Defined 'EOS' as Earth Observing System.

Line 115. Suggest change 'MLS scans the limb…' to 'MLS scans the atmospheric limb…'.

Changed the sentence as follows:

MLS scans the atmospheric limb in the direction of orbital motion which gives almost pole-to-pole coverage (82∘S to 82∘N), leading to retrieved profiles at the same latitude every orbit, with a spacing of 1.5° great circle angle along the suborbital track.

Line 116. '…spaced 1.5 degrees…'. 1.5° in latitude or 1.5° in longitude?

Changed the sentence as follows:

MLS scans the atmospheric limb in the direction of orbital motion which gives almost pole-to-pole coverage (82∘S to 82∘N), leading to retrieved profiles at the same latitude every orbit, with a spacing of 1.5° great circle angle along the suborbital track.

Line 122. Change 'two times a day' to 'twice a day'.

Changed the sentence as follows:

It passes at Ny-Ålesund twice a day at around 04:00 and 10:00 UTC.

Line 122. 'location is within ±400 km latitude and ±800 km longitude'. The units of latitude and longitude should be degrees (°). Either give the ± latitude and longitude ranges in ° or change wording to indicate ± km north–south and ± km east–west of Ny-Ålesund.

Changed the sentence as follows:

Profiles for comparison are extracted if the location is within ±1.2∘ latitude and ±6∘ longitude of Ny-Ålesund and the defined virtual conjugate latitude station.

2.4 MERRA-2

• Line 131. 'We use the ozone and water vapor with 72 hybrid-eta levels…'. What doe 'hybrideta' mean? State whether MERRA-2 model level or pressure level datasets were used.

Changed the sentence as follows:

We use the ozone and water vapor with 72 model levels from the surface up to 0.01 hPa and a horizontal resolution of 0.5∘ × 0.625∘.

**3 Time series of ozone and water vapor**

• Line 135. 'GROMOS-C and MIAWARA-C perform highly accurate continuous ozone and water vapor measurements…'. Why are the measurements 'highly accurate'? To back this up, statements on the measurement accuracies of each instrument are needed (see also Major comment 1). Also, although the instruments have the potential to make continuous measurements, the data gaps in the plots (e.g., Figure 2(a)) suggest the measurements are at best 'near continuous'. What are the reasons for the measurement data gaps?

Changed the sentence as follows: In this study, we use ozone and water vapor measurements from our two ground-based microwave radiometers GROMOS-C and MIAWARA-C which are only available to measure at single locations and, thus, are representative of a specific geographic location.

We agree with the measurements are nearly continuous. The measurement data gaps: during winter 2017/2018 GROMOS-C measured CO for about 2 months and during winter 2016/2017 and summer 2019 the spectrometer had a hardware problem.

3.1 Ny-Ålesund, Svalbard (79 °N, 12 °E) in the NH

• Line 151. '…photo-chemically production…' should be '…photochemical production…'.

Changed the sentence as follows:

Stratospheric ozone largely follows the annual cycle of solar irradiation and is produced through the Chapman cycle, in particular, ozone VMR is mainly dominated by photochemical production in the summer months in the Arctic middle atmosphere.

• Lines 161–162. 'During late summer and early autumn, stratospheric ozone decreases rapidly when the vortex passes over Ny-Ålesund.' This is incorrect: - the polar vortex occurs in wintertime, not late summer and early autumn.

Changed the sentence as follows:

During late winter and early spring, stratospheric ozone decreases rapidly when the vortex passes over Ny-Ålesund.

• Line 163. '…the very big values…'. State the values (see also Major comment 3).

Changed the sentence as follows:

GROMOS-C and Aura-MLS present the seasonal tertiary ozone layer at ~0.03–0.02 hPa / 70–75 km in winter months but the tertiary ozone VMRs are very big values above 0.1 hPa in MERRA-2 (as shown in the red shading in Fig. 2b)

• Line 185. '…water vapor VMR reaches the maximum at 10 hPa.' The 10 hPa pressure level is below the recommended retrieval measurement response (MR) □ 0.8 so how well can these values be trusted? Presumably where MR is below 0.8 the radiometer retrievals are driven primarily by the a priori profile. The a priori data used in retrievals from each instrument needs to be stated.

Stated the sentence as follows:

Radiometer retrievals are primarily driven by the a priori profile. In our study, we used a specific a priori dataset for the retrievals from each instrument. An a priori water vapor profile is required for optimal estimation and is taken from an MLS climatology of the years 2004-2008. We will include a statement in the revised manuscript specifying the a priori data for retrievals from each instrument.

• Line 186. 'In some years,…'. State the years.

Changed the sentence as follows:

In some years such as 2020, water vapor exhibits a larger variability in late winter and spring.

• Line 192. 'MERRA2' should be 'MERRA-2' for consistency.

Changed the sentence as follows:

MERRA-2 shows a tendency to lower water vapor VMR compared to MIAWARA-C and MLS observations.

**3.2 Conjugate latitude station (79 °S, 12 °E) in the SH**

• Lines 198–199. 'For comparison purposes, the conjugate latitude station results are lagged by 6 months relative to those for Ny-Ålesund.' The SH plots in Figs 4-9 are not seasonally lagged but it is recommended that is done (see Major comment 2).

We agree that Figs 4-9 have been replotted and adjusted the date and month on the axes.

• Line 200. '…ozone VMR maximum of about 6 ppmv in winter and a minimum of about 4 ppmv in summer at about 5 hPa…'. The words 'summer' and 'winter' need to be swapped round.

Changed the sentence as follows:

Both stations exhibit annual cycles of ozone while the conjugate latitude station shows an ozone VMR maximum of about 6 ppmv in summer and a minimum of about 4 ppmv in winter at about 5 hPa(Fig. A1b).

• Line 207. '…models…' should be '…MERRA-2 data…'.

Changed the sentence as follows:

This process is well-reflected in the observations and MERRA-2 data for September/October at the conjugate latitude station.

• Line 211. '…and irreversibly remove water and nitric acid from the upper atmosphere…'. Unclear what the upper atmosphere (i.e., thermosphere and above) has to do with this: - suggest remove the words 'from the upper atmosphere'.

Changed the sentence as follows:

Furthermore, polar stratospheric cloud particles can sediment with considerable velocities and irreversibly remove water and nitric acid resulting in a substantial reduction of water vapor VMR at the lower stratosphere during the southern polar winter and early spring (Waibel et al., 1999; Tritscher et al., 2021).

**4 Climatologies of ozone and water vapor**

• Line 216. 'It is important…' Why is it important?

Added the sentence as follows:

It is important to evaluate how well GROMOS-C and MIAWARA-C can monitor the ozone and water vapor variability and enhance our understanding of their distribution in the Arctic middle atmosphere. The inherent variability of ozone and water vapor in the middle atmosphere can be displayed better in the resulting climatologies, as measured by the two instruments involved in this study, and provide a crucial source of data for future work including intercomparison studies and model evaluation, assessing ozone depletion, and validating satellite observations, and studying climate change.

4.1 Ozone

• Line 227. '…maximum observed and modeled in the SH (approximately 5.5 ppmv) is somewhat smaller and earlier in the season…' Suggest change 'maximum observed and modeled' to 'maximum observed and reanalysis ozone VMR'. Also, smaller by how much, earlier by how much, which season?

Changed the sentence as follows:

The maximum observed and reanalysis ozone VMR in the SH (approximately 5.5 ppmv) is somewhat smaller 1.0 ppmv and later in the hemispheric spring season compared to the maximum occurring in the northern hemisphere.

• Line 232. 'Polar cap' usually means above the Antarctic land mass or at very high latitude. Suggest change to 'polar region'.

Changed the sentence as follows:

Precisely, this circulation moves the ozone-rich air from the tropical photochemical source region to high latitude after the polar vortex broke down and essentially enables the intrusion of ozone rich-air from the mid-latitudes into the polar region and replacement of the ozone-depleted air masses.

• Line 236. 'The SH…' should be 'The SH ozone VMR…'

Changed the sentence as follows:

The SH ozone VMR reaches two minimum values in April and August and peaks in the summer and winter. Both hemispheres have dramatically different seasonal variations and distribution in ozone due to differences in the stratospheric dynamics of the two hemispheres.

4.2 Water vapor

• Line 249. '…relatively long photo-chemical lifetime of water vapor, …'. What is the lifetime?

The photochemical lifetime of water vapor in the middle atmosphere is a function of altitude along with the time constants associated with transport by the winds and vertical mixing. The production of water vapor by methane oxidation is essentially complete by about 50 km. Because the photochemical and vertical transport lifetimes for this gas are comparable above about 50 km, and because there is no known chemical source of water vapor in this region, it provides an excellent tracer for mesospheric transport processes (Allen et al., 1981; Bevilacqua et al., 1983; Le Texier et al., 1988).

• Line 251. '…monthly water vapor VMR…' should probably be '…monthly mean water vapor VMR…'.

Changed the sentence as follows:

The annual cycle of monthly mean water vapor VMR at three separate pressure levels (3, 0.3, and 0.02 hPa) in both hemispheres can be seen in Fig. 9

4.3 Relative differences

• Line 271–279. It would be helpful in the text discussion for this section to refer to the specific panels of Fig. 10 e.g., Fig. 10(a) on Line 271 and Figs 10(b-d) at the relevant places for these panels.

We agree that the specific panels are used to discuss this section. Changed the sentence as follows:

Fig. 6a, the largest negative RD is larger than 50% above 0.2 hPa because of the mesospheric ozone parameterization being disabled in MERRA-2 (Knowland et al., 2022).

**5 Dynamics and transport of water vapor**

• Line 300. '…regression fit to different water vapor mixing ratio isopleths…'. State all the isopleth values used here.

Changed the sentence as follows:

With the northern and southern hemispheric water vapor measurements, we calculate the effective ascent and descent rates as derived from a linear regression fit to different water vapor mixing ratio isopleths (5.5, 6.0, and 6.5 ppmv). For instance, the ascent and descent rates from 6.5 ppmv water vapor isopleth are shown in Fig. 11 and Fig. 12).

• Line 301. 'The time period of 7 years…' For MIAWARA-C the time period appears to be ~six years (2015–2020).

Changed the sentence as follows:

The time periods of many years of descent rate from 15 September to 31 October in the altitude range of about 50-70 km is well presented in Fig. 11 (the first and third rows).

• Line 303. '…dynamic and chemical…'. Dynamic and chemical what?

Changed the sentence as follows:

This is an estimated result, not a quantitative calculation of the water vapor descent since the water vapor dynamic and chemicals reactions not directly related to descent may also affect the polar water vapor changes.

• Line 309. '…the ascent rate in 2019 and 2020 is slightly earlier…' Rather than 'slightly earlier', quantify how much earlier.

Changed the sentence as follows:

The starting time of the ascent rate in 2019 and 2020 is about three weeks earlier than that in other years, which is caused by several processes.

• Line 317. Suggest change '…similar for the 7 years, likely due to the higher stability and stronger of the southern polar vortex.' to '…similar for each of the seven years, likely due to the higher stability and strength of the southern polar vortex.'

Changed the sentence as follows:

The descent rate of water vapor from 15 March to 30 April in the SH appears to be similar for each of the seven years, likely due to the higher stability and strength of the southern polar vortex.

• Lines 322–325 and Table 1 (also Discussion and Conclusions sections, also see Major comment 1). The vertical velocities are given to three significant figures, but I wonder whether the linear regression fits are that significant. Would two significant figures suffice? Also, the values only need to be presented once, preferably in Table 1, and discussed in the text. Do the vertical velocities from fits to the different isopleths differ?

The vertical velocities should be given to two significant digits (For example, velocity is ~4.6±0.2 mm/s) and the linear regression fits are very significant. Please see Figure 13, the uncertainties for the ascent and descent rates of air over northern and southern polar latitude stations are depicted by error bars in Figure 13. The vertical velocities fit from different isopleths (5.5ppmv, 6.0ppmv, and 6.5ppmv) and show different isopleths corresponding to different colors in Figure 13.

6 Discussion

• Lines 350. '…Aura-MLS observations…'. Clarify whether the Aura-MLS temperatures are zonally averaged. How much colder are the southern polar latitudes?

Changed the sentence as follows:

In Fig 14, we present zonally averaged temperature at two conjugate latitudes comparison results between the polar and mid-latitudes from Aura-MLS observations in both hemispheres. The southern polar latitudes appear much colder about 20 K than their northern counterparts (Fig B1).

• Line 354. 'There are only a few occasions…' When were these occasions?

Changed the sentence as follows:

There are only a few occasions from 2015 where in the northern hemisphere such a stable and cold polar vortex was observed in 2015/2016 and 2019/2020 in the Arctic winter(Matthias et al., 2016; Lawrence et al., 2020a), which also can lead to anomalies in the middle atmospheric dynamics (Stober et al., 2017).

• Lines 363–364. 'During this time of the year, horizontal gradients…' What time of year is being referred to here? Horizontal gradient in what?

Changed the sentence as follows:

During this summertime of the year, the horizontal gradients of temperature between the polar region and the mid-latitudes are minimal or negligible and, thus, there is no strong forcing due to horizontal temperature gradients or planetary waves driving the zonal or meridional transport.

• Line 380. '…hydroxyl OH…' should be '…hydroxyl (OH)…'.

Changed the sentence as follows:

Later, such a correlation was also found between mesospheric hydroxyl (OH) temperatures and stratospheric reanalysis (Espy et al., 2011).

7 Conclusions

• Lines 388 and 401. 'Ny-Åesund' should be 'Ny-Ålesund'.

Changed the sentence as follows:

In this study, we present ozone and water vapor measurements from the two ground-based radiometers GROMOS-C and MIAWARA-C located at Ny-Ålesund, Svalbard collected between 2015 and 2021.

• Line 389. 'AURA' should be 'Aura'.

Changed the sentence as follows:

The data were compared to observations from MLS onboard the Aura spacecraft as well as reanalysis data MERRA-2.

• Line 393. 'satellite observations such as MLS…' should be 'satellite instruments such as MLS…'

Changed the sentence as follows:

Ground-based observations are going to become more important within the next years as the satellite instruments such as MLS are going to reach the end of their life and so far there are no adequate replacements in orbit.

• Lines 407 and 409. Mention of the quasi-biennial oscillation (QBO) and climate change, and their relevance to this study, should probably be moved to the Introduction.

QBO is moved to the Introduction. Please see the revised manuscript.

---

## Referee Report (RR1)

**Journal:** egusphere

**Title:** Ozone and water vapor variability in the polar middle atmosphere observed with ground-based microwave radiometers

**Author(s):** Guochun Shi et al.

**MS No.:** egusphere-2023-149

**MS type:** Research article

**Special Issue:** Atmospheric ozone and related species in the early 2020s: latest results and trends (ACP/AMT inter-journal SI)

**Review**

The revised paper (version 2) is much improved and largely addresses my comments on the initial submission.  In their response, the authors have addressed my three major comments satisfactorily, although the ozone and water vapour VMR profile uncertainties from the MWR measurements need to be stated in the paper (see minor comments below).  I have several further minor comments and suggestions for improved clarity, and trust that other grammatical and typographical errors will be picked up and corrected at the typesetting stage if the manuscript is approved for publication.

In conclusion, I recommend a further, minor revision before the paper is considered further for publication in *egusphere.*

**Minor comments**

**Abstract**

- Lines 3–4.  The sentence ending ', we analyze the interannual behavior and differences of ozone and water vapor and compile climatologies of both trace gases that describe the annual variation of ozone and water vapor at polar latitudes' could be more succinctly written e.g., ', we analyze the interannual behavior and differences of ozone and water vapor and compile climatologies describing the annual variation of both trace gases at polar latitudes'.

- Line 8.  'MIAWARA-C shows the best agreement with Aura-MLS on average within 5%'.  It should be made clear that the average 5% agreement is between MIAWARA-C and Aura-MLS VMR values.

- Line 15.  '05 May to 20 Jun 2015, …' should be '05 May to 20 Jun in 2015, …'.

**1  Introduction**

- Lines 49–50.  'The quasi-biennial oscillation (QBO) implicit meridional circulation mechanism (Garfinkel et al., 2012) and play an important role…' doesn't make sense and needs to be rewritten.

- Line 78.  'Laser Absorption Spectrometers' should probably be all lowercase i.e., 'laser absorption spectrometers'.

- Lines 79–83.  The sentences 'The ground-based microwave radiometer (MWR) allows a continuous observation under all weather conditions with a time resolution of the order of hours except during rain. It is specially designed for measuring ozone and water vapor which is valuable as it complements satellite measurements, is relatively easy to maintain, and has a long lifetime which ensures a long and continous time series covering several decades, and is operated from different locations and measured autonomously on a campaign basis (Scheiben et al., 2013, 2014). Ground-based microwave radiometry…' could be better written e.g., 'Ground-based microwave radiometers (MWRs) allow continuous observations under all weather conditions with time resolution of the order of hours except during rain. MWRs

measuring ozone and water vapor are valuable as they complement satellite measurements, are relatively easy to maintain, have long lifetimes which ensure long and continous time series covering several decades, and can be operated from different locations with measurements performed autonomously on a campaign basis (Scheiben et al., 2013, 2014). Ground-based microwave radiometry…'

**2.1 GROMOS-C**

- The GROMOS-C ozone VMR uncertainties need to be stated in this section.

**2.2 MIAWARA-C**

- The MIAWARA-C water vapour VMR uncertainties need to be stated in this section.

- Line 125–126. '…orthomode transducer (OMT) placed immediately after the feedhorn. The signal is split into two polarizations by an orthomode transducer directly.' can be shortened to '…orthomode transducer (OMT) located immediately after the feedhorn.' The second sentence isn't needed.

- Line 129. 'Every 15 minutes the ambient load is measured for about 2 s and the sky at 60° elevation is measured for about 15 s.' This suggests only 2 s + 15 s = 17 s of measurements are made every 15 minutes. Confirm this is correct or rewrite to clarify the actual measurement times.

- Line 134. 'MIAWAR-C' should be 'MIAWARA-C'.

**2.3 Aura-MLS**

- Line 143. '118 GHz and 240 GHz radiometers' should probably be '118 GHz and 240 GHz channels'.

- Lines 150–151. 'Profiles for comparison are extracted if the location is within ±1.2° latitude and ±6° longitude of Ny-Ålesund and the defined virtual conjugate latitude station.' should be 'Profiles for comparison are extracted if their location is within ±1.2° latitude and ±6° longitude of either Ny-Ålesund or the defined virtual conjugate latitude station.'

**3.1 Ozone**

- Lines 188–189. The sentence 'The maximum observed and reanalysis ozone VMR in the SH (approximately 5.5 ppmv) is somewhat smaller 1.0 ppmv and later in the hemispheric spring season compared to the maximum occurring in the northern hemisphere' could be written more clearly e.g., 'The maximum observed and reanalysis ozone VMR (approximately 5.5 ppmv) in the SH is 1.0 ppmv smaller than the NH maximum and occurs later in the hemispheric spring season'.

**4.2 Conjugate latitude station (79° S, 12° E) in the SH**

- Line 349. 'MW' should be 'MWR'.

**7 Conclusions**

- Line 479. 'Quasi-Biennial Oscillation' can probably be abbreviated here to 'QBO'.

---

## Author Response (AR2)

**Response to Reviewer – 'Ozone and water vapor variability in the polar middle atmosphere observed with ground-based microwave radiometers' by Guochun Shi et al.**

We thank the Reviewer for his/her positive feedback and valuable comments. Here we address the comments of the Reviewer, with his/her comments in black and our responses indicated in blue.

**Review**

The revised paper (version 2) is much improved and largely addresses my comments on the initial submission. In their response, the authors have addressed my three major comments satisfactorily, although the ozone and water vapor VMR profile uncertainties from the MWR measurements need to be stated in the paper (see minor comments below). I have several further minor comments and suggestions for improved clarity and trust that other grammatical and typographical errors will be picked up and corrected at the typesetting stage if the manuscript is approved for publication.

In conclusion, I recommend a further, minor revision before the paper is considered further for publication in egusphere.

**Minor comments**

**Abstract**

• Lines 3–4. The sentence ending ', we analyze the interannual behavior and differences of ozone and water vapor and compile climatologies of both trace gases that describe the annual variation of ozone and water vapor at polar latitudes' could be more succinctly written e.g., ', we analyze the interannual behavior and differences of ozone and water vapor and compile climatologies describing the annual variation of both trace gases at polar latitudes'.

Changed the sentence as follows:

we analyze the interannual behavior and differences of ozone and water vapor and compile climatologies describing the annual variation of both trace gases at polar latitudes.

Line 8. 'MIAWARA-C shows the best agreement with Aura-MLS on average within 5%'. It should be made clear that the average 5% agreement is between MIAWARA-C and Aura-MLS water vapor VMR values.

Changed the sentence as follows:

The average 5% agreement is between MIAWARA-C and Aura-MLS VMR values.

• Line 15. '05 May to 20 Jun 2015, …' should be '05 May to 20 Jun in 2015, …'.

Added 'in ' in this sentence.

**1 Introduction**

• Lines 49–50. 'The quasi-biennial oscillation (QBO) implicit meridional circulation mechanism (Garfinkel et al., 2012) and play an important role…' doesn't make sense and needs to be rewritten.

Changed the sentence as follows:

The stratospheric quasi-biennial oscillation (QBO) modulates the Northern Hemisphere wintertime stratospheric polar vortex, resulting in its weakening and shifting (Garfinkel et al., 2012; Zhang et al., 2019).

• Line 78. 'Laser Absorption Spectrometers' should probably be all lowercase i.e., 'laser absorption spectrometers'.

Changed the sentence as follows:

Precise water vapor measurements above the troposphere can also be collected by in-situ balloon-borne sensors such as laser absorption spectrometers (Graf et al., 2021)

• Lines 79–83. The sentences 'The ground-based microwave radiometer (MWR) allows a continuous observation under all weather conditions with a time resolution of the order of hours except during rain. It is specially designed for measuring ozone and water vapor which is valuable as it complements satellite measurements, is relatively easy to maintain, and has a long lifetime which ensures a long and continous time series covering several decades, and is operated from different locations and measured autonomously on a campaign basis (Scheiben et al., 2013, 2014). Ground-based microwave radiometry…' could be better written e.g., 'Ground-based microwave radiometers (MWRs) allow continuous observations under all weather conditions with time resolution of the order of hours except during rain. MWRs measuring ozone and water vapor are valuable as they complement satellite measurements, are relatively easy to maintain, have long lifetimes which ensure long and continous time series covering several decades, and can be operated from different locations with measurements performed autonomously on a campaign basis (Scheiben et al., 2013, 2014). Ground-based microwave radiometry…'

Changed the sentence as follows:

Ground-based microwave radiometers (MWRs) allow continuous observations under all weather conditions with time resolution of the order of hours except during rain. MWRs measuring ozone and water vapor are valuable as they complement satellite measurements, are relatively easy to maintain, have long lifetimes which ensure long and continous time series covering several decades, and can be operated from different locations with measurements performed autonomously on a campaign basis (Scheiben et al., 2013, 2014).

**2.1 GROMOS-C**

• The GROMOS-C ozone VMR uncertainties need to be stated in this section.

Added this sentence in 134 line as follows:

The averaging kernels (AVKs) of GROMOS-C together with its measurement response and errors are shown in Appendix A (Fig. A1). In the lower stratosphere the errors are below 0.3 ppmv and reach above the stratopause values up to 0.4 ppmv. More details about the uncertainty and the AVKs can be found in Fernández et al. (2015).

**2.2 MIAWARA-C**

• The MIAWARA-C water vapour VMR uncertainties need to be stated in this section.

Added this sentence as follows:

The AVKs of MIAWARA-C together with its measurement response and errors are shown in Appendix A (Fig. A2). In the upper stratosphere, the errors are 0.5 ppmv and increase from 0.5 ppmv to 1.5 ppmv in the mesosphere.

• Line 125–126. '…orthomode transducer (OMT) placed immediately after the feedhorn. The signal is split into two polarizations by an orthomode transducer directly.' can be shortened to '…orthomode transducer (OMT) located immediately after the feedhorn.' The second sentence isn't needed.

Removed the second sentence and changed the sentence as follows:

The incident radiation is split into vertical and horizontal polarisation by an orthomode transducer (OMT) located immediately after the feedhorn.

• Line 129. 'Every 15 minutes the ambient load is measured for about 2 s and the sky at 60° elevation is measured for about 15 s.' This suggests only 2 s + 15 s = 17 s of measurements are made every 15 minutes. Confirm this is correct or rewrite to clarify the actual measurement times.

Changed the sentence as follows:

The standard measurement cycle of MIAWARA-C is to measure sky East, reference East, sky West, and reference West for about 15 s each. Every 15 minutes the ambient load is measured for about 2 s and the sky at 60° elevation is measured for about 15 s. A tipping curve is performed to determine the sky temperature at 60° elevation. The difference spectra in the east and west directions and the two polarizations are then calibrated separately with the hot and cold measurements close in time.

• Line 134. 'MIAWAR-C' should be 'MIAWARA-C'.

Changed the sentence as follows:

For MIAWARA-C retrievals with a constant time resolution and with a constant noise of 0.014K are performed.

**2.3 Aura-MLS**

• Line 143. '118 GHz and 240 GHz radiometers' should probably be '118 GHz and 240 GHz channels'.

Changed the sentence as follows:

Temperature is derived from radiances measured from the 118 GHz and 240 GHz channels with a vertical resolution between 3 and 6 km.

• Lines 150–151. 'Profiles for comparison are extracted if the location is within ±1.2° latitude and ±6° longitude of Ny-Ålesund and the defined virtual conjugate latitude station.' should be 'Profiles for comparison are extracted if their location is within ±1.2° latitude and ±6° longitude of either Ny-Ålesund or the defined virtual conjugate latitude station.'

Changed the sentence as follows:

Profiles for comparison are extracted if their location is within ±1.2° latitude and ±6° longitude of either Ny-Ålesund or the defined virtual conjugate latitude station.

**3.1 Ozone**

• Lines 188–189. The sentence 'The maximum observed and reanalysis ozone VMR in the SH (approximately 5.5 ppmv) is somewhat smaller 1.0 ppmv and later in the hemispheric spring season compared to the maximum occurring in the northern hemisphere' could be written more clearly e.g., 'The maximum observed and reanalysis ozone VMR (approximately 5.5 ppmv) in the SH is 1.0 ppmv smaller than the NH maximum and occurs later in the hemispheric spring season'.

Changed the sentence as follows: The maximum observed and reanalysis ozone VMR (approximately 5.5 ppmv) in the SH is 1.0 ppmv smaller than the NH maximum and occurs later in the hemispheric spring season.

**4.2 Conjugate latitude station (79° S, 12° E) in the SH**

• Line 349. 'MW' should be 'MWR'.

Changed the sentence as follows: It further demonstrates that ozone and water vapor in the NH will continue to be available from ground-based MWR observations, however, the detailed information about the SH winter as shown here for the 'conjugate latitude' will be lost after the end of the last three limb sounders such as Aura-MLS which is still observing.

**7 Conclusions**

• Line 479. 'Quasi-Biennial Oscillation' can probably be abbreviated here to 'QBO'.

Changed the sentence as follows: However, there is no strong correlation between up- and downwellings in the opposite hemispheres most likely due to dynamic processes such as the QBO or weather patterns which play a role and need to be taken into account.